

# Revisiting Seismic Hazard Assessment For Peninsular Malaysia Using Deterministic And Probabilistic Approaches

Daniel Weijie Loi[1], Mavinakere Eshwaraiah Raghunandan[1], Varghese Swamy[1]

[1]School of Engineering, Monash University Malaysia, 47500 Bandar Sunway, Malaysia

*Correspondence to*: Mavinakere E. Raghunandan (mavinakere.raghunandan@monash.edu)

**Abstract.** Seismic hazard assessments – both deterministic and probabilistic, for Peninsular Malaysia have been carried out using peak ground acceleration (PGA) data recorded between 2004 and 2016 by the Malaysian Meteorological Department – using triaxial accelerometers placed at 19 seismic stations within the peninsula and monitored. Seismicity source modelling for the deterministic seismic hazard assessment (DSHA) used historical point sources whereas in the probabilistic (PSHA)

approach, line and areal sources were used. The earthquake sources comprised the Sumatran Subduction Zone (SSZ), Sumatran Fault Zone (SFZ), and local intraplate (LI) faults. Gutenberg-Richter law b-value for the various zones identified within the SSZ ranged between 0.56 and 1.06 (mean = 0.83) and that for the zones within SFZ, between 0.53 and 1.13 (mean = 0.84). Suitable ground motion prediction equations (GMPEs) for Peninsular Malaysia along with other pertinent information were used for constructing a logic tree for PSHA of the region. The DSHA "critical-worst" scenario suggests PGAs of 0.07-0.80

15   ms$^{-2}$, whilst the PSHA suggests mean PGAs of 0.06-0.42 ms$^{-2}$ and 0.12-0.70 ms$^{-2}$ at 10% and 2% probability of exceedance in 50 years, respectively. Both DSHA and PSHA, despite using different source models and methodologies, conclude that the central-western cities of Peninsular Malaysia located between 2°N and 4°N are most susceptible to high PGAs due to neighbouring active Sumatran sources SFZ and SSZ. Surprisingly, the relatively less active SFZ source with low magnitude seismicity appeared as the major contributor, due to its close proximity. Potential hazard due to SSZ mega-earthquakes should

not be dismissed, however. Finally, DSHA performed using the limited intraplate seismic data from the Bukit Tinggi (LI) fault at a reasonable M$_w$ 5.0 predicted a PGA of ~0.40 ms$^{-2}$ at Kuala Lumpur.

## 1       Introduction

Seismic hazard assessment (SHA) of a particular region can generally be defined as the estimation of hazard at a specific site due to occurrence of a hypothetically damaging earthquake originating within the geographic region. The ground shaking

experienced at a given site is directly related to the intensity of seismic waves emitted by this natural phenomenon. Violent ground shaking caused by devastating earthquakes can lead to both massive fatalities and economic losses, as reported for past earthquake events such as the 2004 Aceh earthquake, 2011 Christchurch earthquake, 2015 Nepal, and 2016 Italy earthquake. The ground motions are normally expressed through response parameters such as peak ground acceleration (PGA), peak ground velocity (PGV), and response spectrum amplitude (RSA). An understanding of the ground motion is one of the





fundamental understanding required to develop reliable seismic resistance design codes that vary from country to country. These design codes established from the ground motion information of a specific region are valuable for practicing engineers in the design of earthquake resistant structures. The SHA methods instigated by strong ground motions have been elaborated in the literature (Baker, 2008; Kolathayar et al., 2012; Kramer, 1996; McGuire, 2001; Panza et al., 1999) with the most common

methods being deterministic and probabilistic approaches.

Deterministic seismic hazard assessment typically uses earthquake magnitude and distance associated with the highest hazard from historical records for a specific seismic source to predict the ground motion at a site. This is commonly achieved using a pre-determined seismic wave attenuation model also known as ground motion prediction equation (GMPE). This method can be termed as a "scenario-like description" for earthquake hazard (Reiter, 1991). DSHA is often desirable for

regions with well-defined seismotectonic models, for example, California, where DSHA dictates the design ground motion for bridges and buildings (Wang, 2011). The application of this approach is straightforward and less complicated allowing engineers to make clear-cut decisions, for consideration of other earthquake parameters unrelated to the site is seldom required. However, DSHA has its own shortcomings in that it does not take uncertainties (i.e., frequency of recurrence and ground motion) into proper account (Baker, 2008; Kramer, 1996). This has inevitably led to the development of probabilistic seismic

hazard assessment (PSHA) which resolves some of the inadequacies in DSHA including probability of recurrence and earthquake magnitude uncertainty.

The use of PSHA has gained popularity in the past two to three decades with the expansion of seismic networks throughout the world and consequent availability of abundant seismic data. The method of PSHA was pioneered by Cornell (1968) and further enhanced by a number of researchers including Esteva (1969), Reiter (1991), McGuire (2004), and Atkinson

et al. (2014). In contrast to the straightforward DSHA method which uses a single absolute value to estimate hazard at a site, PSHA allows the inclusion of multi-valued parameters that consider uncertainties in earthquake factors such as the location, size, and the recurrence rate. The combination of these parameters provides an advantage for SHA as it enables assessment of the likelihood of an earthquake ground motion exceeding a certain threshold at a site of interest. PSHA employs flexible mathematical approaches which are oftentimes presented in the ground motion annual return rate of exceedance or return

period which facilitates engineers to perform seismic risk assessment for a site of interest. Subsequently, with better understanding of the seismic hazard, specifically on the relationship between different sources and the potential shaking caused by impending earthquakes, engineers can ascertain suitable design ground motion that a structure should be able to withstand. This method, nonetheless, is not free of criticism as studies have observed that PSHA is merely a numerical creation with a hazy mathematical concept and the use of it may lead to risky or overly conservative engineering design (Klügel, 2010; Wang,

2011). It is always a good practice, therefore, to supplement the results from PSHA with analyses conducted using DSHA.

As Malaysia is a developing nation with new infrastructure being built at a relatively fast rate in her major cities, it is essential that seismic hazard assessment is undertaken to reliably predict ground motion scenarios due to potential earthquakes. The ground motion values obtained will serve as a reference for upcoming constructions and also for existing structures as an evaluation to determine if retrofitting is required to mitigate the seismic risk. Currently, the design code BS8110 is widely



used by the construction industry in Malaysia and the ongoing usage of this design code can be deemed unwise as it does not include any seismic considerations (Megawati et al., 2005, Shoustari et.al, 2016) It is worth noting that the inherent seismic hazard for the Malaysia region has been acknowledged by the Government of Malaysia. In view of the lessons learnt from the devastating earthquakes of the Sumatran region, especially in the aftermath of the 2004 Aceh Earthquake, there have been

initiatives such as publication of a handbook on the requirement of incorporating seismic design, in particular for concrete buildings in Malaysia based on Eurocode 8 and IBC2000 design codes (Ministry Of Science Technology & Innovation, 2009). However, the values proposed in these codes may not be suitable for usage as they were not specifically developed for this region (Sooria, 2012). The reason being that the seismotectonic parameters such as earthquake sizes and frequency, distance from the sources, among others, vary for different regions throughout the world.

In view of both its methodological limitation in not treating uncertainties adequately and that ground motions felt within Peninsular Malaysia have been predominantly due to infrequent distant events, the utilization of DSHA in Peninsular Malaysia has been relatively scarce. Unsurprisingly, PSHA has been the choice for SHA by a number of researchers in this region. The PSHA outcomes reported for this region have been recently discussed by Loi et al. (2016) and Shoushtari et al. (2016). These authors have discussed possible reasons for the variation in the published PSHA outcomes including the

utilization of different GMPEs and datasets (either synthetic or recorded ground motions), employment of different methodologies for PSHA, and site-specific conditions.

The major motivation for the current study is the lack of a dedicated GMPE for Peninsular Malaysia. The past studies adopted regional GMPEs not specifically developed for Peninsular Malaysia for SHA of this region. Moreover, awareness of potential earthquake hazards in the country has gained traction over the last decade owing to a series of minor earthquakes in

Bukit Tinggi between 2009 and 2010 and the Sabah Earthquake in 2015. In 2016, the Department Of Standards Malaysia (2016) also drafted an Annex – denoted as DMS16 in this paper – based on Eurocode 8 on the applicability of seismic resistant design in Malaysia. With intensifying interest in earthquake studies in Malaysia, the present work aims to contribute a detailed study of the seismic hazard faced by Peninsular Malaysia including the development of seismic zonation maps. To this end, updated strong ground motion records obtained from the Malaysian Meteorological Department (MMD) for the period of

2004-2016 in conjunction with the recent findings on the suitability of existing and new GMPEs for this region (Loi et al., 2018; Shoushtari et al., 2016; Van et al., 2016) will be used in performing DSHA and PSHA for Peninsular Malaysia encompassing a rectangular area of 1ºN to 7ºN and 99ºE to 105ºE. The outcomes of the present research comprises (a) seismic hazard maps based on both DSHA and PSHA via ground motion in terms of PGA at bedrock and (b) hazard curves for major cities throughout the Peninsula. The PSHA hazard map will also present the PGA with 2% and 10% probabilities of exceedance

(PE) in 50 years.



## 2      Tectonic setting and seismicity of Peninsular Malaysia

The foremost step in the SHA for a region is the identification of the potential earthquake sources capable of yielding substantial ground motion at a given site. The earthquake sources vary from active interplate subduction regions where earthquake recurrence is relatively high as the result of constant interactions between tectonic plates to stable continental intraplate regions which are away from the plate boundaries and can be identified based on historical seismological events and geological data. The knowledge of the seismotectonic setting of a region is derived on the basis of past seismicity and geological structures. The area considered in the present study consists of the whole of Peninsular Malaysia located between the latitudes 1º N and 7º N and longitudes 98º E and 105º E (Fig. 1).

Peninsular Malaysia that covers an area about 0.3 million km2 lies at the southern tip of mainland Asia and is connected by land to Thailand to the north while it is separated from Singapore by Johor Straits to the south and from Sumatra Island of Indonesia by Malacca Straits to the west. The Borneo Island which comprises the states of Sabah and Sarawak, on the other hand, is located east of Peninsular Malaysia and is separated by South China Sea. Tectonically, Peninsular Malaysia is located within the stable Sunda Plate. Seismicity within the Sunda Plate has been historically low with progressive collision with the Eurasian Plate relatively slow (Baroux et al., 1998). The axis of rotation of the Sunda block is believed to be at 49.0ºN to 94.2ºE with a clockwise rotation of 0.34 degree/million years (Simons et al., 2007). The general movement of this block is eastwards at a slow- rate of $6 \pm 1$ mm/y and $10 \pm 1$ mm/y in its southernmost and northern boundaries, respectively (Simons et al., 2007). Despite being located on a stable continental region, ground motions due to earthquakes (both major and minor) are still being experienced within the country (Megawati et al., 2005; Ministry Of Science Technology & Innovation, 2009; Sun and Pan, 1995). Based on the chronological events documented by various agencies such as the United States Geological Survey (USGS), International Seismological Centre (ISC) and MMD, it could be established that ground motions detected due to seismic activity within and around Peninsular Malaysia can largely be attributed to two main sources: farfield Sumatran sources and local intraplate earthquakes. These two sources can further be grouped into three seismotectonic regions: Sumatran Subduction Zone (SSZ), Sumatran Fault Zone (SFZ), and intraplate zones within the Sunda Plate. Historical statistics obtained from MMD showed that states located on the western coastline of Peninsular Malaysia are more vulnerable to felt ground motions (Loi et al., 2016; Sooria, 2012). The location of Peninsular Malaysia within the Sunda Plate and its nearby seismic sources are presented in Fig. 1.

## 3      Interplate faults in the Sumatran region

Figure 2 schematically illustrates the tectonic movements around the Sumatran region that lead to major seismic activities. The island of Sumatra located on the Eurasian Plate overrides the subducting Indian-Australian plate along the Sunda Trench. The subduction zone which lies on the Indian Ocean bed is not as distinctive as the fault lines on Sumatra. This zone, where the two plates converge has generally been identified as the Sumatra Subduction Zone. The SSZ is relatively younger south of the equator (approx. 50 Ma) and older towards the north (approx. 90 Ma) with historical records showing that earthquakes of



high magnitudes happening frequently at younger and faster moving subducting plates (Cassidy, 2015; Gradstein et al., 1994; Gutscher, 2016). This does not imply that mega earthquakes are not likely to happen at zones that are moving at a slower convergence; the 2004 Aceh earthquake being a prominent example of the latter (McCaffrey, 2009). The convergence of these plates is highly oblique to the southwest of Sumatra, lying almost parallel and approximately 150 – 200 km away from its coastline. The vector of plate motion varies around $57 \pm 8$ mm/y and is oriented about N10°E (McCaffrey, 1991; Megawati et al., 2005; Petersen et al., 2004; Prawirodirdjo et al., 2010). The resultant mega earthquakes are directly related to the strong coupling between the overriding and subducting plates with studies indicating that the focal mechanism and hypocentral distribution being shallow and dips gradually beneath the outer arc ridge (Newcomb and McCann, 1987; Pan and Megawati, 2002; Prawirodirdjo et al., 1997). SSZ has accounted for most of the megathrust earthquakes in this region with records showing one of the largest earthquakes ever to strike had a massive $9.0 \pm 0.2$ on the moment magnitude ($M_w$) scale in 1833 (Newcomb and McCann, 1987). Another massive earthquake happened in 1861 at an estimated $M_w$ of $8.4 \pm 0.1$, which was felt in Java and Peninsular Malaysia (Newcomb and McCann, 1987). More recently, the Aceh Earthquake recorded at ~ $M_w$ 9.1-9.3 near the island of Simulue (Nalbant et al., 2005), generated giant tsunamis that lead to thousands of fatalities and posed colossal financial losses, in terms of rebuilding and restoration work to the surrounding regions. Although high rise buildings were not structurally damaged in distant countries such as Malaysia and Singapore, tremors were still reportedly strongly felt even as far as India (Martin, 2005).

Lying dextral and parallel about 200 km away from the trench to accommodate the oblique convergence along the plate margin is the Sumatran Fault Zone. This 1900 km long strike-slip fault runs in a NW-SE direction along the spine of Sumatra, spanning from 10°N to 7°S (Sieh and Natawidjaja, 2000). The slip rate of this fault accelerates northwestwards at varying speeds of 6 to 27 mm/yr with relatively high seismicity rates in the vicinity of Sumani, Sianok and Angkola (Petersen et al., 2004; Prawirodirdjo et al., 2000). This is in line with the Global Positioning System (GPS) data studied by McCaffrey et al. (2000) that suggested a uniform slip rate of $21 \pm 5$mm/year across central Sumatra. A geomorphology study on the SFZ by Sieh and Natawidjaja (2000) and Acocella et al. (2018), found that it is highly segmented and can be divided into 19 major geometrically defined segments. Termed "equatorial bifurcation', the largest irregularity is located at the equator, where the fault separates into two subparellel branches at approximately 35 km apart (Sieh and Natawidjaja, 2000). The geometrical irregularities exhibited along the sinusoidal shape of Sumatran Faults have tectonic and seismological significance that affects the rupture dimensions, limiting the energy that could be released from this active strike-slip fault (Balendra et al., 2002). This is supported by historical data noting that major earthquakes in this zone have never exceeded $M_w$ 7.8 (Natawidjaja and Triyoso, 2007). The same study also concluded on the basis of the assumption that all the fault zones are locked from surface to a depth of 15 km, that the recurrence of large earthquake $M_w$ $7.2 – 7.4$ is approximately 0.2/year while an earthquake of $M_w$ 7.4-7.7 is likely to strike 0.1/year,. Although earthquakes from SFZ are comparatively lower in magnitude when compared to those from the SSZ, the effects during major ruptures such as the 2010 and 2011 events were still being felt in Peninsular Malaysia. The logical explanation is that the lower magnitude effect of the earthquakes from SFZ is offset by the shorter distance to the peninsula.



## 4 Intraplate faults within Peninsular Malaysia

The geological map published by the Mineral and Geoscience Department of Malaysia (JMG) recognizes three prominent set of fault systems trending in NW-SE, N-S and E-W directions. Seven major faults were listed within the Peninsular Malaysia, including Bok Bak fault, Lebir fault, Terengganu fault, Bukit Tinggi fault, Kuala Lumpur fault, Lepar fault, and Mersing fault

(Mineral and Geoscience Department Malaysia, 2014). From November 2007 to May 2008, a series of low magnitude ($M_w$ <4.0) earthquakes were registered at Bukit Tinggi. These events were sufficient to generate tremors felt by nearby residents and minor hairline cracks were spotted on the wall at a nearby police station and school (Lat and Tajuddin, 2009; Lau et al., 2005). Such occurrences were unanticipated as seismicity level within Peninsular Malaysia has historically been low with observed intensities of level VI on the Modified Mercalli (MM) scale due to tremors instigated by Sumatran events (Chai et

al., 2011). These events presumably suffice after the megathrust earthquakes at Aceh and Nias in 2004 and 2005, respectively, with recent geophysical studies suggesting that the core of Sundaland to be gradually deforming (Shuib, 2009). This notion is supported by GPS and Shuttle Radar Topography Mission – Digital Elevation Mapping (SRTM-DEM) measurements showing distortion of plates due to intraplate stress build up in the northwest of Peninsular Malaysia (Jhonny, 2009). Such movements seemingly activate the intraplate faults, eventually leading to low magnitude intraplate earthquakes. Considering that Kuala

Lumpur (KL), the capital of the nation, is located only about 30 km away, these events warrant general public's interest and concern. The presence of these local intraplate (LI) earthquakes requires further geomorphological studies for a better understanding of the faults' behavior and level of seismicity these faults are capable of producing. A new hazard map incorporating potential hazards posed by these active faults will certainly be useful for engineers as a provision during seismic resistant design stage.

## 5 Earthquake database and catalogue

Over the past 15 years, the Malaysian Meteorological Department (MMD) has set up a network of seismic stations across Peninsular Malaysia. In view of economic and scientific importance, majority of these stations are located in the west coast of Peninsular Malaysia where major cities are situated. Moreover, they are located closer to the active Sumatran region. The network comprises of 19 stations that use FBA-EST triaxial accelerometers. Out of these 19 stations, 7 are equipped with

broadband seismometers (Streckeisen STS-1 and STS-2). The sensors used at these stations by MMD capture the horizontal, vertical, and surface accelerations due to an earthquake event. Real time data are transmitted via VSAT telemetry to the headquarters of MMD for processing and analysis. As these stations were built on various foundations namely, granite, sandstone, and soft soil, the sites are referenced to the National Earthquake Hazards Reduction Program (NEHRP) site classification by the Building Seismic Safety Council (2003). The aforementioned two foundations on which 13 seismic

stations have been founded on can be classified as NEHRP site class B – rock sites (average shear wave velocity in the upper 30m (VS30) of the soil profile with VS30 ranging from 760 to 1500 ms-1); whereas the later soft soil where 5 seismic stations were founded is considered to be NEHRP site class E (VS30 ranging from 760 to 1500 ms-1). The data from one remaining





seismic station located within a building were not considered in the current study. The details of these stations (location, foundation, NEHRP site class and recorded PGA ranges) are listed in Table 1.

For the period of 2004 to 2016, a total of 88 earthquake events within a rectangular area of 10°S to 10°N and 95°E to 110°E that triggered considerable ground motion were recorded by the MMD. The data set for PGA consists of 103 recordings for local earthquakes and 368 recordings from farfield Sumatran earthquakes. 34 out of 88 events were categorized as low magnitude local earthquakes which occurred within Peninsular Malaysia and are of $M_w \leq 4.0$ whereas the remaining 54 earthquakes were classified as farfield earthquakes from the SSZ and SF. These events were located more than 400 km away and have recordings ranging from $M_w$ 5.0 – 9.1. The focal depth of LI earthquakes ranges from the surface to a depth of 22.5 km while the focal depths for far field earthquake range from 9 km to 580.9 km. PGA data utilized in this study were from the original uncorrected accelerograms and were not post-processed as they are normally smaller due to time decimation and frequency band-limited filtering (Campbell, 1981). Since the recorded PGA values (in vertical and two perpendicular horizontal directions) across Peninsular Malaysia were very low (0.00003 to 0.0616 ms$^{-2}$), the peak value from an individual recording was utilized as the worst case scenario in this study. 378 records were from rock sites (NEHRP class site B) while the remaining were from soil sites (NEHRP class site E).

Apart from the recorded data collected from MMD, the information of past earthquakes around the Sumatran region were obtained through the USGS and ISC earthquake catalogues as a comprehensive SHA analysis requires a sizeable amount of data. The combined catalogue comprises earthquake data for the region 10°N - 7°S and 90°E - 106°E with minimum earthquake magnitude of $M_w \geq 4.0$ for the period of 4th January 1907 to 31st December 2016. The total events in the raw catalogue were 22,734. However, considering that earthquake hazard is usually estimated using a Poisson model, not all data from the catalogue were suitable as they contained both foreshocks and aftershocks. The "de-clustering" (removal of the dependent events i.e., foreshocks and aftershocks from background seismicity) leads to a better estimation of random events which is a vital aim in SHA (Kolathayar and Sitharam, 2012). For this purpose, the de-clustering was performed using the algorithm proposed by Gardner and Knopoff (1974). This process, together with the removal of duplicates, eliminated 18,286 dependent events with the remaining 4,448 events identified as main shocks. Out of these 4,448 events, 1,414 events were from SFZ with $M_w \geq 4.0$ and the remaining 2,954 were from SSZ with $M_w \geq 5.0$.

## 6    Source modelling

Identification of the seismic source model based on geological evidence, geotectonic province, historic seismicity, geomorphic investigation, and other relevant data is one of the crucial steps in SHA. For the present study the earthquake sources utilized to define the source models have been confined to an area encompassing 91°- 106°E and 10°N - 7°S. Here the assumption is that earthquakes that are capable of causing significant ground motion originate as far as approximately 800 km radius away from the most northwestern point of Peninsular Malaysia - the island of Langkawi - and the southernmost point – considered Singapore here.





DSHA oftentimes presents the worst-case scenario of an earthquake event and consideration of the probability of location and time of occurrence plays a less critical role compared to PSHA (Moratto et al., 2007). Although ground motion data collection only began since 2004 in Peninsular Malaysia, records of great earthquakes ($M_w$ >8.0) from the Sumatra region are available for the period since 1797 (Newcomb and McCann, 1987). It would be insightful to model these historical events

also to predict the PGA values across Peninsular Malaysia. For this purpose, point sources instead of line and areal sources are utilized here to replicate the historical events. With no clear segmentation for the SSZ, as opposed to the SFZ, a grid of 1.0° x 1.0° and a limitation of 200 km on either side of the digitized subduction line were considered to cover the entire area. The maximum possible earthquake (MPE) utilized for the analyses was the largest earthquakes with $M_w$ ≥7.0 that occurred within the same grid in the past. In addition, a simulated event of $M_w$ 9.1 was presumed at the Mentawai-Siberut segment (2°S, 99°E)

as studies have reported the possibility of a mega earthquake within the next couple of decades (Lay, 2015; Philibosian et al., 2014). On the other hand, the fault lines on the SFZ have been researched more extensively and are better wedged compared to the SSZ, with 19 segments spanning across mainland Sumatra, as listed in Sieh and Natawidjaja (2000). Therefore, events with $M_w$ ≥6.0 along these segments were considered as the MPEs. As for the local intraplate events, although a few major faults have been identified within the peninsula, only minor earthquakes from Bok Bak and Bukit Tinggi faults have produced

notable ground motion, and therefore, only 6 events with magnitude $M_w$ >2.4 were considered.

With the MPEs thus determined, the next step was to assign a maximum possible magnitude to these locations. Multiple scenarios were considered for this objective. Scenario 1 represents the maximum historical earthquake recorded by ISC, USGS and also Newcomb and McCann (1987) for the Sumatran region while the maximum magnitudes for local earthquakes were recorded by the MMD. Earthquake magnitudes that were recorded in body-wave magnitudes (Mb) especially

for the data collected from MMD were converted to $M_w$ using the regression suggested in Loi et al. (2018). As it is almost impossible to determine if past events will be superseded by earthquake of larger magnitude, one standard rule of thumb that has been employed to consider the "worst-case" scenario is to increase the magnitude of past events by $M_w$ 0.25 or 0.5 (Naik and Choudhury, 2014; Secanell et al., 2008; Shukla and Choudhury, 2012). Hence, this method was assigned to Scenario 2 whereby due to its slower convergence, an increment of 0.3 $M_w$ was applied to events originating above the equator from the

SSZ. In addition, this zone has undergone massive rupture, frequently releasing strain energy in recent times which have resulted in mega earthquakes of $M_w$ 8.6, 8.6, and 9.0 in year 2005, 2012 and 2004, respectively. On the other hand, an increment of $M_w$ 0.5 was applied to events located below the equator from the same region due to this region's faster convergence and also because researchers have predicted that a major earthquake may happen along the Mentawai segment within the next few decades (Lay, 2015; Nalbant et al., 2005). The maximum magnitude applied was, however, limited to $M_w$ 9.5 considering that

the largest ever earthquake recorded was the $M_w$ 9.5 1960 Chilean earthquake. Similarly, an increment of $M_w$ 0.5 was assigned for events emanating from the SFZ with a maximum magnitude of $M_w$ 8.0. Within the peninsula, records for the local intraplate events have been scarce and sporadic. Hence, the MPEs for the local intraplate events were retained as per Scenario 1 as it is difficult to estimate now a credible maximum magnitude for the faults. Nonetheless, taking into account that KL lies in close proximity to three major fault lines (Bukit Tinggi, Seremban, and KL Faults) and records indicating that stable continental



earthquakes have the odd capability of striking above $M_w$ 6.0 (Johnston and Kanter, 1990; Schulte and Mooney, 2005), a plausible increment of $M_w$ 1.0 was assigned to the Bukit Tinggi event. The values from Scenarios 3 and 4, by contrast, were obtained from literature and are only applicable for the SFZ. Scenario 3 tabulates the predicted maximum magnitude for each of the 19 segments with a 200 year return period by Natawidjaja and Triyoso (2007), while Scenario 4 represents the predicted

maximum magnitude for each of the 16 tessellated zones in SFZ using k-means algorithm analytical approach by Burton and Hall (2014). The maximum magnitudes for each of the four scenarios were thereafter compared with the largest value being utilized as the MPE.

A total of 50 MPEs were identified from all three regions (SSZ, SFZ and LI). 25 events were for the SSZ with the largest anticipated events coming from the 2004 Aceh earthquake and the simulated Mentawai-Siberut earthquake at $M_w$ 9.4

and $M_w$ 9.5, respectively, while smaller events ($M_w$ of 7.3 –7.8) were projected around the Nicobar Islands cluster between 6°N and 9°N. The least maximum magnitude for the SFZ was located near the Toru, Baruman and Manna segment, recorded at $M_w$ 6.0 while the largest was from the Sumani segment, recorded at $M_w$ 7.8. Despite the relatively low magnitudes recorded for the former, Natawidjaja and Triyoso (2007) estimated based on rate of seismic moment calculation that a maximum magnitude for these three segments may be as high as $M_w$ 7.4. The maximum magnitude calculated by Burton and Hall (2014)

for the same zones was even higher, in the range of $M_w = 7.6 - 7.8$. The maximum magnitude estimated by these two literature sources was noticeably higher when compared to actual recordings and, therefore, were selected as the MPEs for our DSHA. As for the local earthquake scene, the highest MPE utilized for DSHA was that of the Bukit Tinggi earthquake. A detailed list of these events from all three regions with four different scenarios and the selected MPEs is presented in Table 2 and the locations are illustrated in Fig. 3.

Similar to DSHA, one of the crucial steps in PSHA is to identify the seismic source model. While DSHA in the current study utilizes point source, linear and areal sources were used for the PSHA. Although the utilization of the latter two sources have been well documented in the literature (Anbazhagan et al., 2008; Kramer, 1996; Ornthammarath et al., 2010; Vipin et al., 2009), specifying the linear and area sources for SSZ is complicated owing to the following: the SSZ is extremely long (over 4000 km), its location off the coast of Sumatra Island, and key tectonic parameters such as its segmentation length,

displacement, and area are not well defined. The subduction line utilized in the PSHA analyses for the SSZ were approximately digitized using the USGS maps. In regard to the upper and lower boundaries of SSZ, past observations have noted that majority of the earthquakes tend to strike at a certain depth to the right of the subduction line, instead of to the left, due to the subduction of the Indian-Australian plate (Fig. 4). This phenomenon is more prominent to the south of the equator as illustrated in Fig. 4. Keeping in mind that large earthquakes are capable of striking on both sides of the subduction line, the boundary width of the

areal source for SSZ was confined to be within 200-250 km on either side of the subduction line and away from Sumatra Island. As the age of the plate and slip rates differ from north to south, this zone was further segmented into 7 different zones at every 2° or 3° latitude intervals with different expected maximum magnitude ($M_wMax$) for each individual zone.

In contrast to SSZ, the occurrences of earthquakes to the left and right of the SFZ are almost equal throughout. Although the SFZ has been better defined, as shown by Natawidjaja and Triyoso (2007), some of the subdivided segments





tend to overlap making the fault line boundary determination somewhat complicated. Therefore, for the latitudinal margin for the SSZ, the boundary was divided as per the suggestion by Burton and Hall (2014). However, the SFZ in the present work is subdivided into 13 instead of 16 segments as suggested by Burton and Hall (2014). This is achieved by combining the southernmost three segments into one segment in view of the fact that these are located relatively far off from the area of our interest. The width of these zones was, however, not uniform: to the left of the fault line the zone width was constrained to be within Sumatra Island while to the right, the width varied from approximately 20 to 100 km away from the fault line. A map of the source modelling zonation for the PSHA is illustrated in Fig. 4.

While multiple scenarios were considered to determine the MPEs in the DSHA, in the PSHA for SSZ the present analysis considers that a $M_w$Max earthquake could take place all along the SSZ even though the values vary from north to south. With slip rates towards the north relatively slower compared to those in the south, the upper boundary $M_w$Max for Zone 1 was fixed at 9.0 with the values gradually increasing until a maximum of 9.5 for Zone 7. By contrast, multiple MPEs for the SFZ from Table 2 fall within a same zone for some cases in the current study. As such, the $M_w$Max is selected based on the highest MPE within the same zone. The length, slip rate, and $M_w$Max for each zone are given in Table 3.

Although not directly related to the PSHA, Table 3 also summarizes the observations for earthquake occurrences per year for the past 40 years (since 1976) for every interval of $M_w$ 1.0 from both zones. This is despite that the SHA considers records from USGS since 1907. The approximate range of 40 years was chosen based solely on observation. The reason is that the records for the years prior to 1976 are relatively scarce. Besides, throughout the years, the expansion of ground motion stations worldwide and collection of earthquake data have progressively increased, and it is difficult to determine a cut-off point to which time should reliable data be considered. Moreover, data prior to 1976 consist of <8% of the overall records, after the removal of foreshocks and aftershocks. The records for the SSZ clearly show that earthquake occurrences in Zone 7 are relatively higher compared to that in Zone 1, in line with studies suggesting movement rates are higher in the south of the SSZ, thereby indicating that higher slip rates result in higher frequency of earthquakes. A similar pattern, however, cannot be observed for the SFZ wherein the earthquake frequency is rather scattered with no clear correlation between the slip rate and the frequency of earthquake occurrences. This is reflected for the SFZ in Zones 1 and 4 wherein although the pair have similar fault lengths but varying slip rates, the frequency of occurrences was still comparable at 0.75 and 0.975, respectively. Similarly, both Zones 8 and 9, despite having analogous fault lengths and slip rates did not result in similar frequency of occurrences. Apart from that there also seems to be no direct link between slip rate and the upper boundaries of $M_w$ for both regions.

## 7        Regional seismicity recurrence

One of the most commonly used methods to characterize seismicity for a region is the Gutenberg-Richter earthquake recurrence law (Gutenberg and Richter, 1944). This law estimates the seismic parameter b-value which follows a magnitude exponential distribution expressed as:

$$log_e\, Nm = a - bM \tag{1}$$





where $Nm$ is the total number of earthquakes exceeding $M$ for the predetermined region, $a$ is a constant that reflects the earthquake productivity or seismic activity, and $b$ indicates the relative occurrence of small and large events. Larger b-values, the slope of frequency versus magnitude distribution (FMD), implies a larger proportion of small earthquakes whereas a small b value represents relatively small number of large magnitude earthquakes (Nanjo et al., 2012). Of the two variables, the b-value has often been prioritized by researchers and has undergone many statistical and analytical evaluations over the past few decades. It has been widely recognized that this value normally hovers around 1.0 for seismically active regions (Baker, 2008; El-Isa and Eaton, 2014; Mogi, 1962; Singh et al., 2015).

A least-squares regression method was utilized to obtain the b-values for the studied region with earthquake threshold magnitude above $M_w$ 5.0 for the SSZ and 4.0 for the SFZ. Figure 5 presents the FMD plots for the SSZ and SFZ as a whole and also for each of the 7 and 13 zones individually with the b-values listed in Table 3. It should, however, be reminded that the b-values in the table has no relation to the observation column in Table 3 as the FMD plots considered data since 1907 and not only for the past 40 years.

As illustrated in Fig. 5, the b-values range between 0.56 and 1.06 for the SSZ and between 0.53 and 1.13 for the SFZ. The estimated b-value for Zone 3 in SSZ was noted to be particularly low as this zone has been associated with only a few earthquakes of with $M_w$ >8.0 since year 2000. As for the SFZ, the estimated low b-values for Zone 1 is due to the moderately short length of Zone 1 with historically large earthquakes ($M_w$ >6.0). The low b-value for Zone 9, in spite of its relatively long length, is due to the comparatively low earthquake recurrences on top of the occurrence of odd earthquakes with high magnitude ($M_w$>7.0). Albeit their relatively low b-values, the average for the overall regions of SSZ and SFZ was higher at 0.83 and 0.84, respectively. These values concur well with the b-values for the PSHA obtained for Sumatra Island and KL by Irsyam et al. (2008) and Nabilah and Balendra (2012). Petersen et al. (2004) performed PSHA for Sumatra, Singapore, and Peninsular Malaysia using proposed b-value of $1.02 \pm 0.16$.

## 8      Ground motion prediction equations (GMPEs)

Suitable GMPEs that can predict/estimate ground motions in good agreement with recorded ground motion data due to past seismic events are fundamental to SHA. Although numerous GMPEs have been developed and applied worldwide, not many GMPEs are available exclusively for Peninsular Malaysia due to its relatively lower local seismicity and distant location from active seismic hotspots such as the Sumatran region. Naturally, past attempts either adapted or adopted regional GMPEs or relied on the available limited data for developing GMPEs suitable for this region (Adnan et al., 2005; Pan and Megawati, 2002; Petersen et al., 2004). The collection of seismic ground motion data since 2004 by MMD, albeit relatively smaller in quantity compared to more earthquake active regions, has since allowed researchers to either identify suitable GMPEs (Van et al., 2016) or develop independent GMPEs for the peninsula (Adnan and Suhaltril, 2009; Loi et al., 2018; Nabilah and Balendra, 2012; Shoushtari et al., 2016) using the available ground motion records. Loi et al. (2016), Van et al. (2016), and Shoushtari et al. (2015) have compared the adaptability of selected worldwide GMPEs revealing their limitations wherein most of them



either overestimated or underestimated the actual ground motion data for the peninsula. Therefore, more accurate GMPEs developed for this region by Loi et al. (2018) and Shoushtari et al. (2016) together with the GMPE developed for Japan by Si and Midorikawa (2000) are used here to carry out the DSHA and PSHA. As for the LI earthquakes, we only carried out DSHA using the Nguyen et al. (2012) GMPE that fits best the scarce data of low magnitude events (Loi et al., 2018). The relationship

of these GMPEs to recorded ground motion data due to the SSZ, SFZ, and LI earthquakes are plotted at various magnitude intervals in Fig. 6.

## 9        Logic tree structure

There are inherent uncertainties associated with earthquake data and these uncertainties can be broken down into two categories: aleatory (statistical) and epistemic (systematic) (Bommer et al., 2005). Whereas aleatory uncertainty is unavoidable

due to the fact that earthquake is a random process, epistemic uncertainly (limited knowledge and data) can be accounted for using a logic tree structure (Bommer et al., 2005; Delavaud et al., 2012; Marzocchi et al., 2015; Youngs and Coppersmith, 1985). A logic tree consists of a series of nodes that lead to multiple branches. The branches allow a formal characterization for addressing uncertainties in the analysis by including parameters and models (hypothesis), each being subjectively weighted on the basis of engineering judgment and their probability of being accurate. The weightage for each individual branch leading

up to the end branch can be multiplied to obtain the weightage of that particular route and the sum the weightages should equal to one. Parameters selected for constructing logic tree formation in this study include different regions, source modelling, magnitude uncertainty model, b-values and GMPEs.

For DSHA, the selected GMPEs from the respective regions were weighted to predict the value of PGA at a site of interest. As two different GMPEs were suggested for SSZ and SFZ in Loi et al. (2018), denotations of SSZL18 and SFZL18

were given to differentiate the models used for SSZ and SFZ respectively in this study. As the SSZL18 showed more robustness compared to S16 (denoted for Shoushtari et al., 2016) for the ground motion data from the SSZ, especially at lower magnitude range (Fig. 6a), weightages of 0.6 and 0.4 were assigned to each GMPE, respectively. On the other hand, weightages were equally split for the GMPEs applicable to the SFZ as both SFZL18 and SM00 (denoted for Si and Midorikawa, 2000) showed close estimation in relation to recorded ground motion data while GMPE suggested by N12 (denoted for Nguyen et al., 2012)

was utilized as the only GMPE for LI earthquakes. An in-house Microsoft Excel based program was designed to perform the DSHA with hazard outcome being the maximum possible PGA estimated as a function of distance and magnitude taking into consideration each of the 50 MPEs.

For PSHA, both the SSZ and SFZ were weighted equally. The reason for this choice is that although the SSZ is capable of producing earthquakes of higher maximum magnitude compared to SFZ, the former is located relatively further

away from Peninsular Malaysia. The SSZ and SFZ were subsequently split into 7 and 13 zones, respectively with each zone comprising two different sources: line and area, weighted equally at 0.5 each. Moving down the branch to address the magnitude uncertainty of each zone, the line sources were branched out into three categories: $M_w$ 6.0 − 7.5, $M_w$ 7.5 − 8.5 and



$M_w$ 8.5 – MAX with weightages of 0.6, 0.3, and 0.1 respectively for the SSZ and $M_w$ 5.0 – 6.5, $M_w$ 6.5 – 7.5 and $M_w$ 7.5 – MAX with identical weightages assigned to the SFZ. The reason for assigning higher weightages to the lower boundaries is that frequency of earthquakes at lower magnitudes are much higher compared to earthquakes with higher magnitude. On the other hand, the area sources for both SSZ and SFZ were assigned full weightages of 1.0 from $M_w$ 6.0 – MAX and $M_w$ 5.0 –

MAX, respectively. The weightages for b-values - separated into fixed (for overall region) and variable (for individual zones) were also equally split. The PSHA was subsequently conducted using the same weightages for the GMPEs as used in the DSHA. A PSHA logic tree structure with the respective weightages to the branches is shown in Fig. 7. PSHA calculations using the input parameters such as source models, b- values, $MaxM_w$, logic tree weightages, and GMPEs were conducted using EZ-Frisk v8.00 developed by Risk Engineering Inc, USA.

While PSHA performs integration on all the possible earthquake occurrence of and ground motions to predict the mean frequency of exceedance, the knowledge of the source relative contribution to the hazard in terms of distance and magnitude is oftentimes valuable and deaggregation is one such method (Bazzurro and Allin Cornell, 1999; McGuire, 1995; Trifunac, 1989). Deaggregation of PGA was carried out in terms of bin pairs of distance and magnitude (R, M) at 20 km and $M_w$ 0.1, respectively, following the procedure presented in EZ-Frisk.

**10   Results and discussion**

**10.1   Hazard maps**

Two cases were considered for this study. Case 1 considered the mean values from the GMPEs to predict the PGA whereas Case 2 considered the mean values from the GMPEs plus their respective upper boundary standard deviation to predict the PGA. It should be noted, however, that for the local intraplate MPEs, only the mean values from N12 were used for both cases

(discussed below). Two separate DSHA maps for Case 1 and Case 2 were subsequently plotted with the hazard values for each grid point using ArcMap 10.4. (Fig. 8a and 8b). Figures 9a and 9b were plotted for SSZ and SFZ individually using Case 2 considering this can be termed as the "critical-worst" case to determine the MPEs that contribute to the ground motion hazard for the major cities across Peninsular Malaysia.

As observed for Case 1 in Figs. 8a and 8b, the PGA value varies from 0.02 to 0.34 ms$^{-2}$ across the peninsula while

the PGA values expectedly rise approximately 2.5 fold for Case 2 in the range of 0.07 - 0.80 ms$^{-2}$. Both Figs. clearly depict that lower central-western part (below latitude 4.0°N) of Peninsular Malaysia is more susceptible to higher seismic hazard with PGA values decreasing from the southwest to northeast of Peninsular Malaysia. When the overall DSHA map is split into the regional sources (SSZ and SFZ), as shown in Figs 9a and 9b, it is observed that the source that contribute to the high PGA in the cities of KL, Seremban and Melaka was from the SFZ with the MPE associated located close to the Angkola segment.

Although this event is noted to occur slightly off the Sumatra fault line compared to the remaining events from the SFZ, this hypothetical MPE further illustrates that the controlling earthquake could be located closer to the peninsula and hence fits in with worst-case scenario often associated with DSHA. Conversely, the high PGA predicted in the northwestern islands of





Penang and Langkawi originates from the SSZ with the MPE associated being the epicenter of the 2004 Aceh Earthquake, hereby modelled at $M_w$ 9.4. It is also worth noting that in spite of having simulated a hypothetical event near the Siberut-Mentawai segment at $M_w$ of 9.5, the PGA estimated at KL, Seremban, and Melaka from this SSZ event was still lower when compared the event originating at Angkola from the SFZ, thereafter highlighting the hazard that the SFZ may produce. Nevertheless, the PGAs predicted at southern peninsula and Singapore from both regions were similar, with the SSZ capable of producing PGA ranging from 0.16 to 0.20 ms$^{-2}$ while the SFZ is expected to produce PGA between 0.18 and 0.24 ms$^{-2}$ at JB and Singapore.

Although there were six MPEs in total associated with the local intraplate earthquakes, only three MPEs were large enough to produce high PGAs compared to the events originating from the Sumatran region (see contour lines in Fig. 8). The remaining three MPEs were of very low magnitude at less than $M_w$ 3.0. Of particular interest is the MPE modelled at $M_w$ 5.0 close to the Bukit Tinggi Fault. In relation to this fault, the PGA predicted within the 20 km vicinity from the centre of KL (3.14°N and 101.69°S) can reach as high as 0.4 ms$^{-2}$ with the value peaking at 0.5 ms$^{-2}$ approximately 10 km away from the epicenter (Fig. 10). Although this work considered $M_w$ 5.0 as a plausible case, concerns have been raised by Looi et al. (2013) in an extreme event of $M_w$ 6.0 which cannot be ruled out. Therefore, utilizing the same source but altering the maximum magnitude to $M_w$ 6.0, the PGA values for this special case was further calculated and plotted in Fig. 10 for comparison purpose. The PGA observed was exceedingly high and the predicted values were capable of raising as high as 3.0 ms$^{-2}$. This value is approximate 6 and 4 times more than the PGA expected from the MPE for the LI events and Case 2 for the Sumatra region, respectively. Although the predicted PGAs show a sharp drop to 1.2 ms$^{-2}$ at the centre of KL, these values are still alarmingly high. This is certainly expected as the Nguyen et al. (2012) GMPE used for the DSHA is applicable up to a suggested local magnitude ML of 4.6. Therefore, until a better understanding of the critical magnitude that these LI faults can produce is achieved coupled with a more suitable GMPE, this value may be too conservative to be implemented for seismic resistant design in KL. Furthermore, seismic resistant design for countries located on stable continental regions with low seismicity worldwide mostly have a threshold at 0.2g (Giardini et al., 2013; Woessner et al., 2015). The PGA values from the current work fall within previous DSHA studies performed by local researchers. Manafizad et al. (2016) predicted the PGA across the country in the range of 0.01 - 0.191ms$^{-2}$ while the estimate by Adnan et al. (2005) was relatively low, between 0.03 and 0.07 ms$^{-2}$

## 10.2    Probability of exceedance (PE) maps and hazard curves

Now considering the PSHA, it has been well established that earthquake designs are based on 10% and 2% PE in 50 years (return period of 475 and 2475 years, respectively) with the outcome expressed in hazard curves and macrozonation contour maps of mean PGA. For the current study, it should be noted that these hazards are calculated based on rock site condition with references to NEHRP class B with the average shear-wave velocity being 760 ms-1 in the upper 30m of the crust.

Figure 11 presents the hazard curves in terms of mean annual rate of exceedance against PGA at various cities across Peninsular Malaysia which clearly highlights that the hazard in central-western cities (between latitudes 2°N and 4°N) being the highest,



followed by the northwestern (above 4°N) and southern (below 2°S) cities (including Singapore). The information from the hazard curves are reflected in the regular PE maps displayed in Fig. 12. The ground motions across Peninsular Malaysia expressed in PGA at bedrock ranges from 0.06 to 0.42 ms$^{-2}$ and 0.12 to 0.70 ms$^{-2}$ for 10% and 2% PE in 50 years, respectively. Although the PGA values differ, both maps exhibit a similar pattern in that the PGA values gradually decrease from southwest

towards northeast of the peninsula. Once again, higher PGA values were observed for KL and Melaka with the lowest PGA estimated at Kuantan. Even though the DSHA indicated that the southern region is more susceptible to higher hazard in comparison to the northwestern region, the PSHA suggest that the hazard at both regions were comparable with PGA values ranging between 0.20 ms$^{-2}$ and 0.25 ms$^{-2}$. The results are further compared with similar PSHA work from the past literature and seismic resistance values suggested in DMS16.

The bars next to the PE maps in Fig. 12 show the PGA ranges estimated across Peninsular Malaysia by various researchers in the past. The PGA estimated from a study by Pan and Megawati (2002) – denoted as PM02 – for 10% and 2% PE in 50 years was between 0.13 and 0.30 ms$^{-2}$ and 0.24 and 0.55 ms$^{-2}$ across Singapore and Peninsular Malaysia, respectively. A separate study conducted by Petersen et al. (2004) – denoted as P04 – predicted relatively high PGA values of 0.40 – 1.17ms$^{-2}$ and 0.78 - 1.96 ms$^{-2}$ while Adnan et al. (2005) – denoted A05 – predicted values between 0.10 and 0.25 ms$^{-2}$ and 0.15 - 0.35

15  ms$^{-2}$ across the peninsula at 10% and 2% PE in 50 years. Another separate study by Adnan et al. (2006) – denoted as A06 – predicted rather high PGA with values ranging from 0.20 to 1.00 ms$^{-2}$ and 0.40 to 2.00 ms$^{-2}$ for the same 10% and 2% PE in 50 years. As for the more recently drafted DMS16, a definitive range was not clearly indicated for the same return periods, but it was suggested that ordinary buildings (defined as low rise structures/individual dwellings) were to be designed against 0.69 ms$^{-2}$ at 10% PE in 50 years while important critical structures such as hospitals, emergency services, power stations and

communication facilities should be designed against 0. 98 ms$^{-2}$ at 2% PE in 50 years. The PGA calculated from this work presents a wider range of hazard across the peninsula when compared to the predictions by A05 and PM02. While the PGA calculated at the higher spectrum coincides with the PGA for the lower range of A06, the PGA data from this study do not agree well with the PGAs calculated by P04. We believe that the current work possibly represents the seismic ground motion experience in the peninsula better than the previous studies given that the earthquake data used here is richer and the GMPEs

applied more reliable in relation to the actual ground motion records.

**Deaggregation and hazard source**

The combined deaggregation results from both regions at 10% and 2% PE in 50 years across the major cities in the peninsula and Singapore are displayed in Fig. 13. The results provide information regarding the magnitude-distance combinations which have major contribution to the PGA values together with the mode and mean distances and magnitudes.

The results show that the SSZ is the main hazard contributor at Langkawi Island, southern region (Johor Bahru and Singapore) and eastern region (Kuantan). Meanwhile, Penang Island, despite being situated relatively close to Langkawi, along with the cities from central-western region (Ipoh, KL and Melaka) are more susceptible to hazards originating from the SFZ,



especially at the 10% PE in 50 years. However, at a longer return period, the source zone that contributes to the higher PGA for all the western cities were noted to originate from the SFZ, except for Kuantan.

Furthermore, hazard sources affecting three major cities representing the north and central regions along the west coast and also Singapore were selected for comparison in Fig. 13. It can be observed that the major source that contributes to
5 the hazard in Penang is the SSZ, in line with results from DSHA. Meanwhile, hazards calculated at KL were likely due to events located in the SSZ for PGA less than 0.42 ms$^{-2}$ while events from SFZ contribute more at higher PGA, albeit at a noticeably lower frequency. A similar trend was also observed for Singapore where the hazard contribution at PGA less than 0.66 ms$^{-2}$ was mainly from the SSZ. The hazard curve for SSZ though gradually tapers towards the hazard curve for SFZ at higher PGA, indicating that hazard posed by SFZ increases at higher PGA.

**Conclusion**

In summary, this paper presents an overall SHA in terms of PGA at bedrock for Peninsular Malaysia using the DSHA and PSHA approaches. Historical point sources were modeled in DSHA while line and areal sources were utilized for PSHA. Earthquake data collected from the literature, ISC, USGS and MMD were utilized for source modelling and the estimation of seismic hazard parameter "b". The b-values for various zones from the SSZ and SFZ range between 0.56 and 1.06 and 0.53-
15 1.13 with mean values of 0.83 and 0.84 respectively using the GR-Law. Suitable GMPEs were subsequently employed with the assistance of a logic-tree structure for the SHA. Both DSHA and PSHA, despite having different seismic source models and conducted using different software (in-house Microsoft Excel based for DSHA and EZ-Frisk for PSHA) conclude that the central-western cities (latitudes 2°N to 4°N) of Peninsular Malaysia are most susceptible to high PGAs due to their location closer to the seismically active Sumatran region. The DSHA using "critical-worst" case indicated that the hazard across
Peninsular Malaysia on bed rock in terms of PGA ranges from 0.07 to 0.80 ms$^{-2}$, while hazard conducted using PSHA at PE for 10% and 2% in 50 years (return periods of 475 and 2475 years, respectively) showed that the mean PGA ranges from 0.06 to 0.42 ms$^{-2}$ and from 0.12 to 0.70 ms$^{-2}$, respectively. Similarly, the combined results from both the SHA showed that the hazard across the peninsula (especially below 5°N latitude) was mostly contributed by the SFZ albeit the latter being less active and the limited energy it releases. However, it is worth mentioning that the current work only focuses on the PGA at bedrock
without taking into consideration the spectral acceleration and soil amplifications. Hence, the contribution of mega earthquakes from the SSZ frequently associated with long duration seismic waves should not be dismissed.

The absence of good seismic data (small database and short duration activities) for the local intraplate events prevented the utilization of PSHA. Nevertheless, a simulated DSHA near the Bukit Tinggi fault at a reasonable M$_w$ 5.0 predicted a PGA of approximately 0.40 ms$^{-2}$ at the center of KL. The overall hazard from both deterministic and probabilistic
analyses, albeit their differences, lead to similar results and offer valuable information on the seismic ground motion experience across the peninsula. Finally, the PGA values from SHA were lower than the recommended values from the drafted Annex on the seismic resistant design from the Department of Standards Malaysia (2016) which was adjusted based on Eurocode 8,



suggesting that the usage of the Annex, for now, is suitable across the peninsula. However, revisiting the SHA procedure with a new set of earthquake data set and improved approaches is recommended in future, which defines the accuracy and reliability of the assessment procedure.

*Acknowledgements.* The authors would like to acknowledge the financial support from the Ministry of Higher Education Malaysia through the Fundamental Research Grant Scheme grant (FRGS/2/2013/TK03/MUSM/03/2). The authors would also like to acknowledge the Malaysian Meteorological Department for providing the earthquake data and details pertaining to their measurement and the seismic network in the nation. The contribution of Dr. Biswajeet Pradhan, Distinguished Professor, University of Technology Sydney (formerly Universiti Putra Malaysia) and Dr. Zainuddin Md. Yusoff, Faculty of Engineering,
Universiti Putra Malaysia in terms of suggestions during initial stages of earthquake data collection is highly appreciated.

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





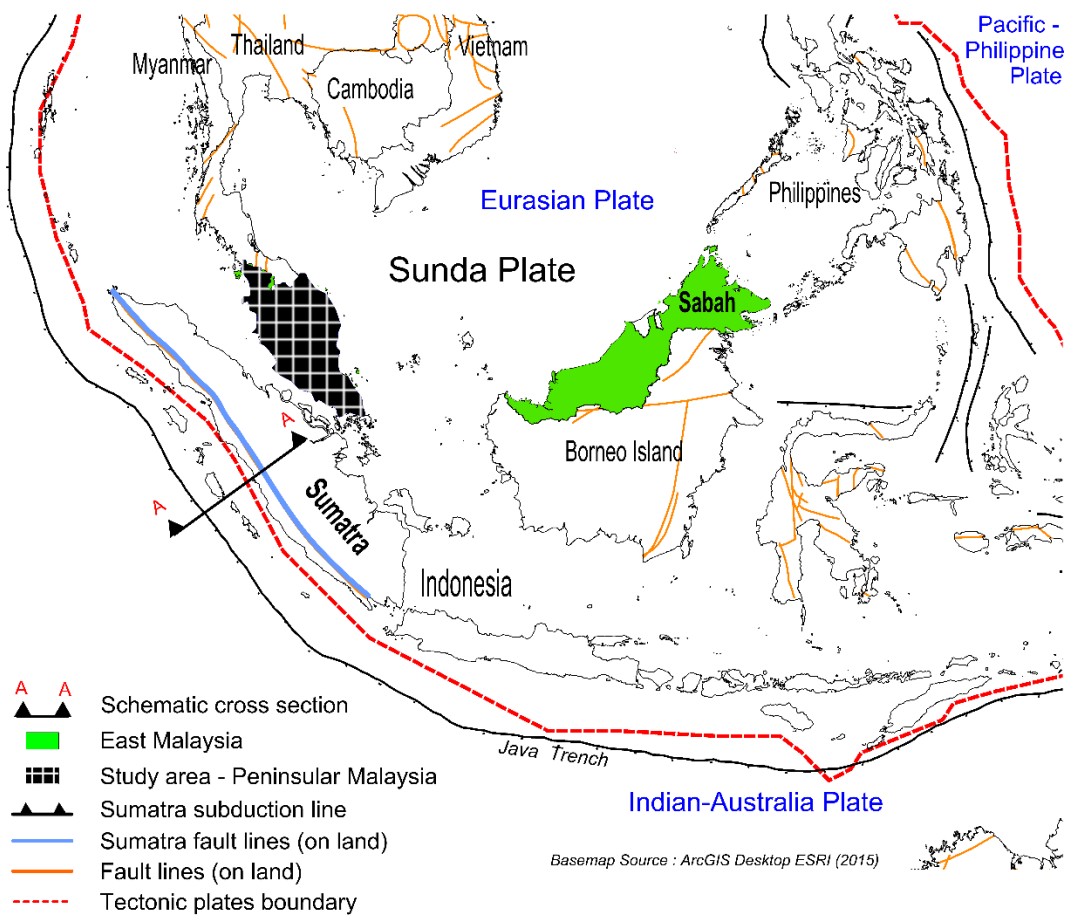

**Figure 1** Location of Peninsular Malaysia on the Sunda Plate and the seismic sources around it (modified after Loi et.al, 2016).





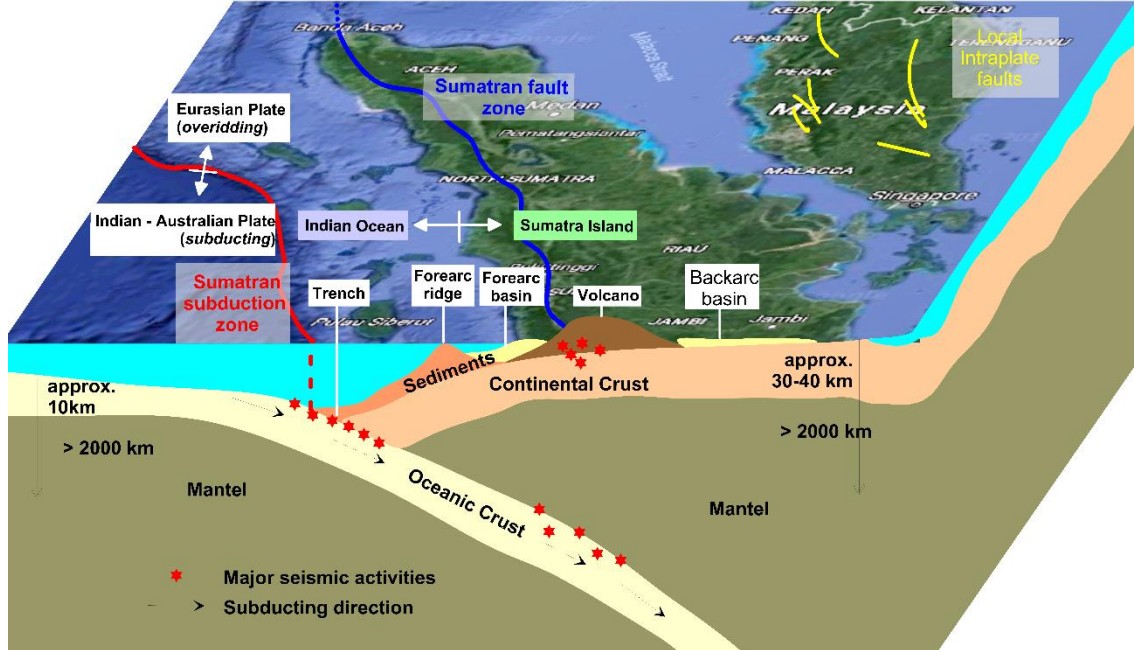

**Figure 2.** Schematic cross-section of A-A from Fig. 1 showing the subduction of Indo-Australian Plate beneath the Eurasian Plate and the location of major seismic activities along the Sumatra subduction and fault zone.



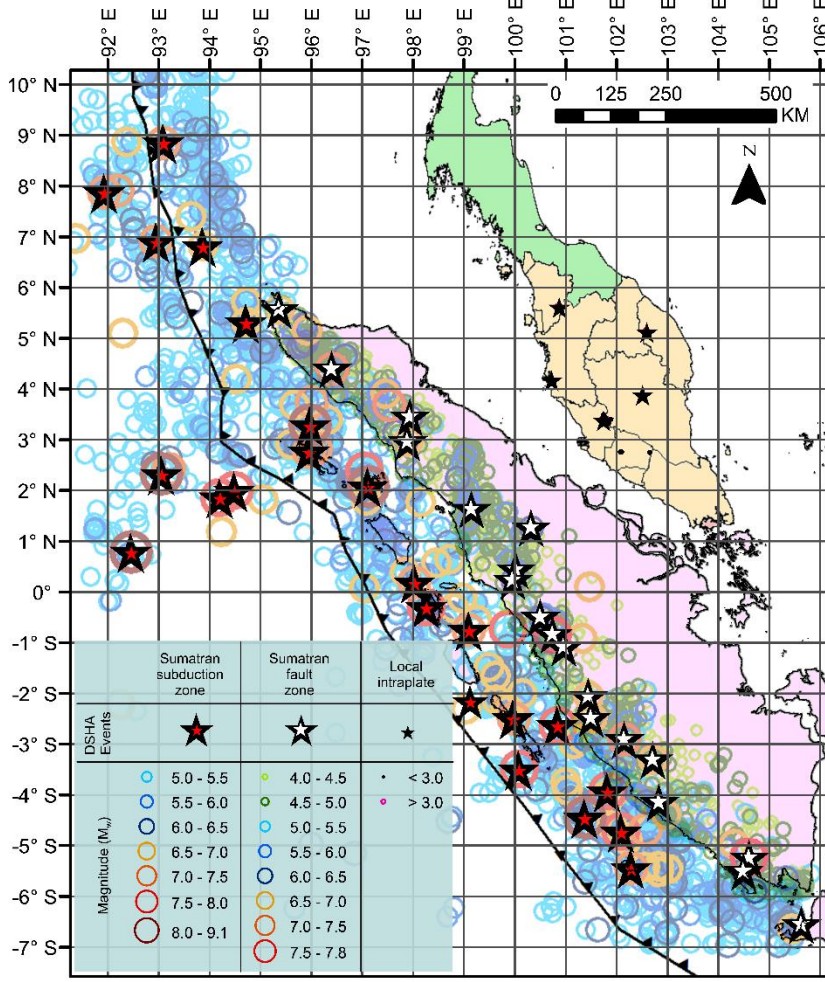

**Figure 3.** An epicenter map of historical earthquake magnitudes $M_w \geq 5.0$ for the Sumatran Subduction Zone, $M_w \geq 4.0$ for the Sumatran Fault Zone, and low magnitude earthquakes within Peninsular Malaysia for the period of 1906 – 2016. The records for these events were taken from USGS earthquake catalogue, MMD, and published literature. Earthquake sizes are given on scales and colors proportional to the earthquake magnitudes. The asterisks show the locations of the MPEs utilized for DSHA for each region.


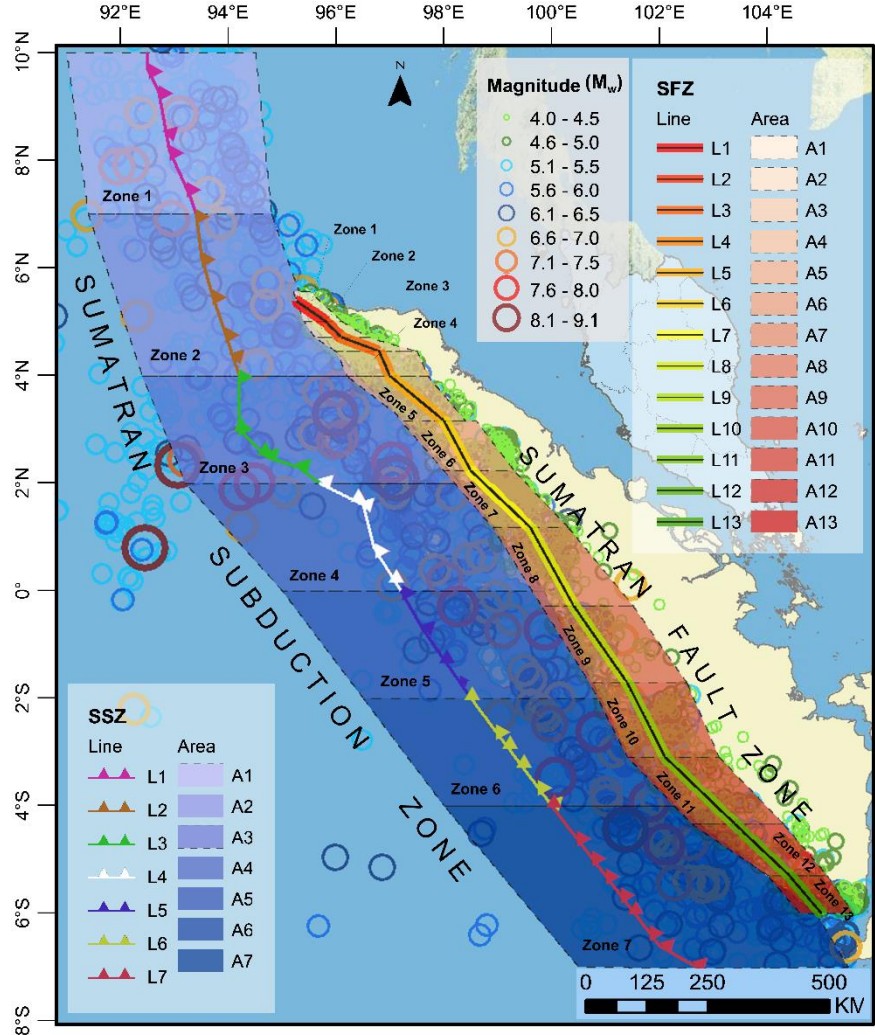

**Figure 4.** Seismic zonation map for the Sumatra regions with SSZ and SFZ being split into two different source models (line and area) for PSHA. The details of these zones such as length, slip rate, and $M_w$Max are listed in Table 3.





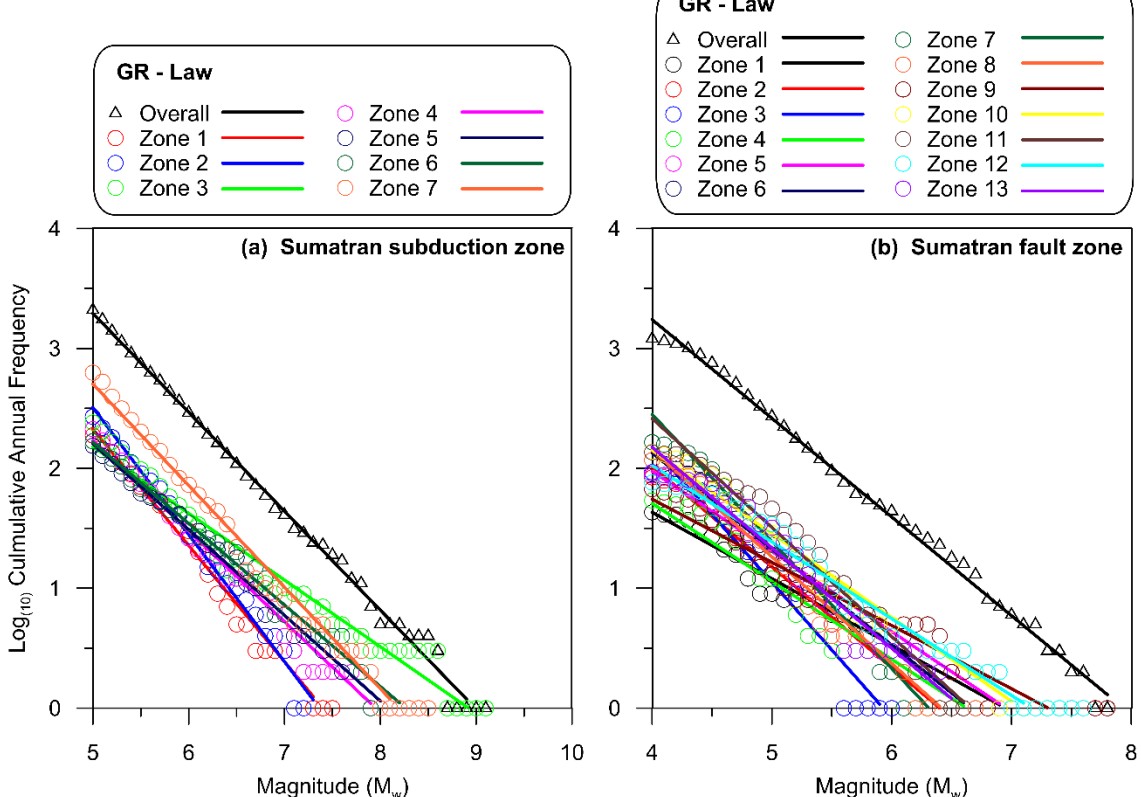

**Figure 5.** Magnitude versus cumulative annual frequency relation obtained using the GR-Law for (**a**) Sumatran subduction zone and (**b**) Sumatran fault zone. The b-values are listed in Table 3.





**Figure 6.** Plots at various magnitude intervals for the GMPEs used in the current study with respect to recorded ground motion data for (**a**) Sumatran Subduction Zone using GMPEs proposed by Loi et al (2018) and Shoustari et al (2016), denoted as SSZL18 and S16, respectively, (**b**) Sumatran Fault Zone using the GMPEs proposed by Loi et al (2018) and Si and Midorikawai (2000), denoted as SFZL18 and SM00, respectively, and (**c**) Local intraplate fault zone using the GMPE proposed by Nguyen et al. (2012), denoted as N12.





**Figure 7.** Logic tree structure with weightages for PSHA.





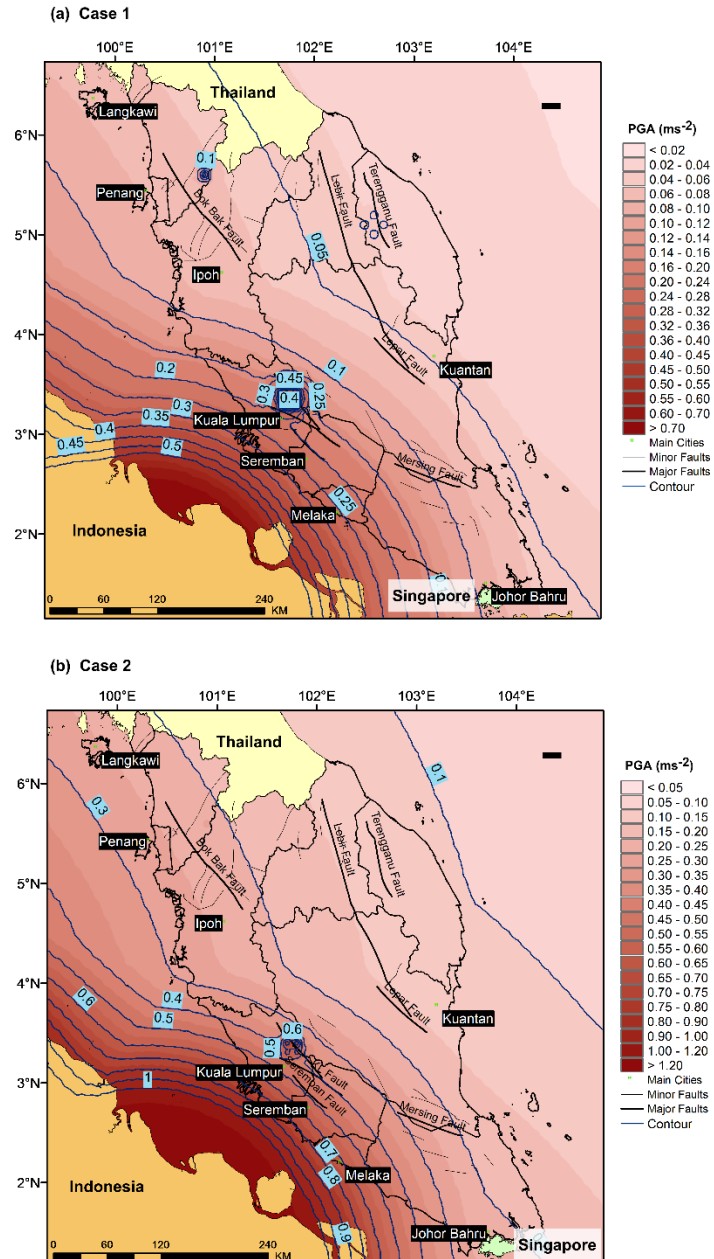

**Figure 8.** PGA maps of Peninsular Malaysia obtained using DSHA. (**a**) Case 1 – mean GMPE, (**b**) Case 2 as "critical – worst" case – mean GMPE plus +ve standard deviation.





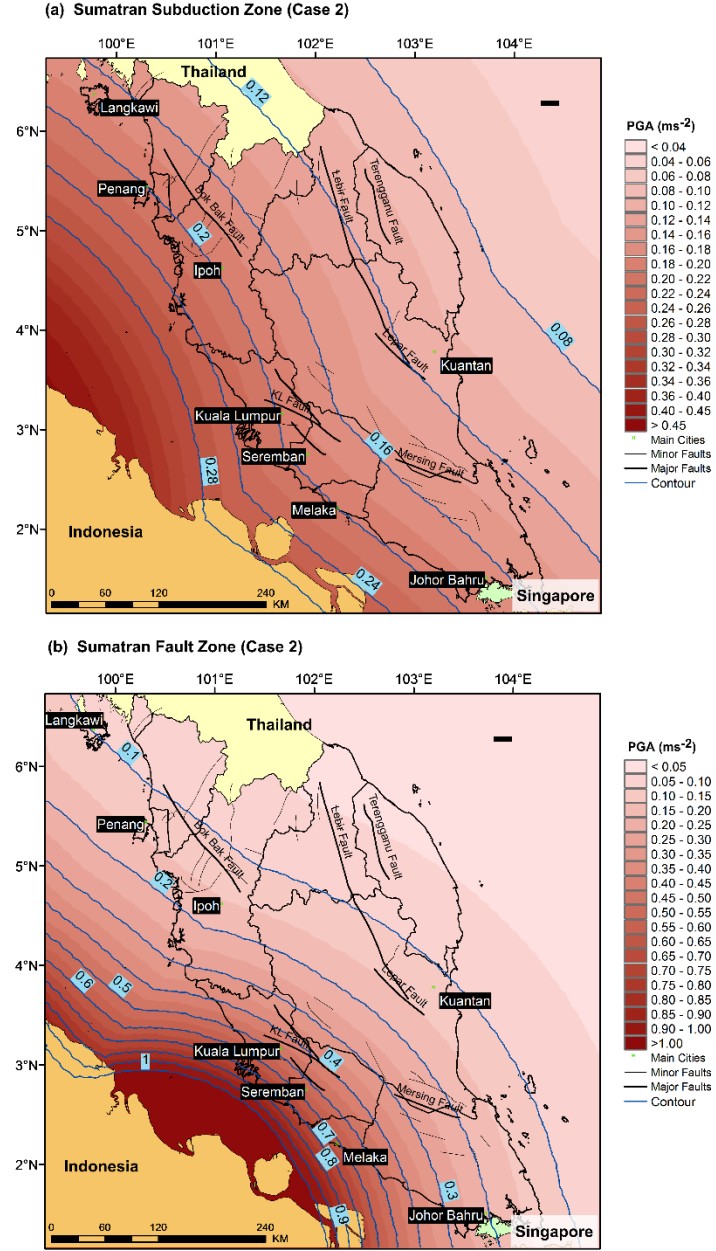

**Figure 9.** PGA maps of Peninsular Malaysia for Case 2 for the sources originating from (**a**) Sumatran Subduction Zone and (**b**) Sumatran Fault Zone based on Case 2.





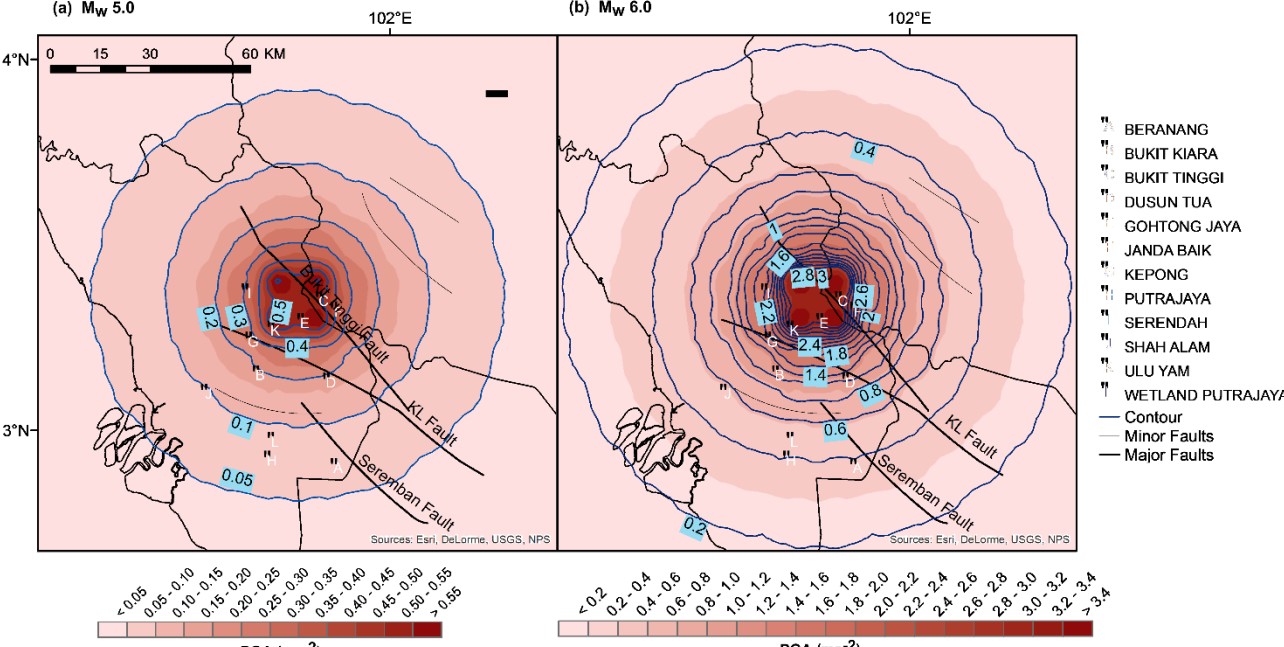

**Figure 10.** PGA map based on a simulated event of $M_w$ 5.0 and 6.0 from the Bukit Tinggi Fault.





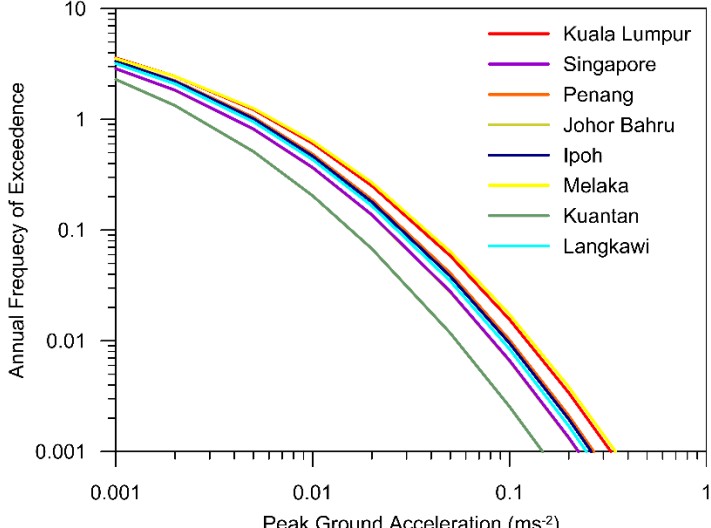

**Figure 11.** Hazard curves for different cities in Peninsular Malaysia and Singapore at rock sites.





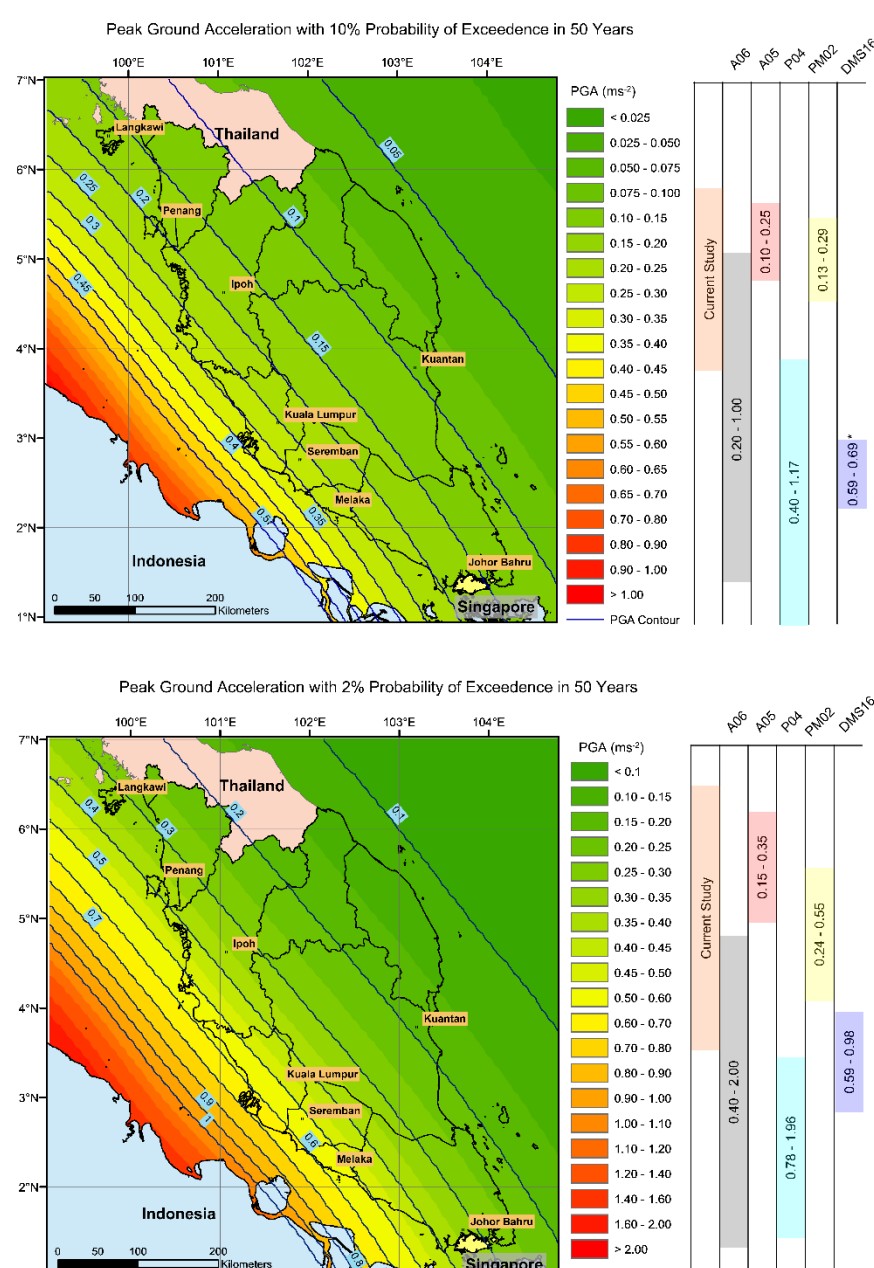

**Figure 12.** PGA Maps of Peninsular Malaysia at rock site condition affected by the Sumatran sources at 10% and 2% in 50 years probability of exceedance respectively.


**10% OVERALL PE**

|  | Amplitude (ms$^{-2}$) | Mode Magnitude (M$_W$) | Mode Distance (km) | Mean Magnitude (M$_W$) | Mean Distance (km) |
|---|---|---|---|---|---|
| Langkawi | 0.173 | 7.45 | 570 | 7.97 | 528 |
| Penang | 0.199 | 7.45 | 350 | 7.88 | 492 |
| Ipoh | 0.193 | 7.45 | 350 | 7.88 | 504 |
| KL | 0.256 | 7.55 | 330 | 7.73 | 429 |
| Melaka | 0.279 | 7.75 | 250 | 7.70 | 308 |
| JB | 0.161 | 7.45 | 630 | 7.97 | 518 |
| Singapore | 0.162 | 7.45 | 630 | 7.96 | 516 |
| Kuantan | 0.098 | 7.45 | 710 | 8.21 | 662 |

**2% OVERALL PE**

|  | Amplitude (ms$^{-2}$) | Mode Magnitude (M$_W$) | Mode Distance (km) | Mean Magnitude (M$_W$) | Mean Distance (km) |
|---|---|---|---|---|---|
| Langkawi | 0.338 | 7.45 | 390 | 8.22 | 506 |
| Penang | 0.388 | 7.45 | 350 | 8.12 | 473 |
| Ipoh | 0.376 | 7.45 | 350 | 8.13 | 486 |
| KL | 0.499 | 7.55 | 330 | 7.92 | 407 |
| Melaka | 0.544 | 7.75 | 250 | 7.85 | 362 |
| JB | 0.319 | 7.75 | 350 | 8.19 | 487 |
| Singapore | 0.320 | 7.75 | 330 | 8.18 | 485 |
| Kuantan | 0.196 | 7.45 | 710 | 8.50 | 636 |

**Figure 13.** Deaggregation plots showing PGA relative contribution from the Sumatran region for Peninsular Malaysia and Singapore as a function of distance and magnitude at various major cities at 10% and 2% PE, respectively.



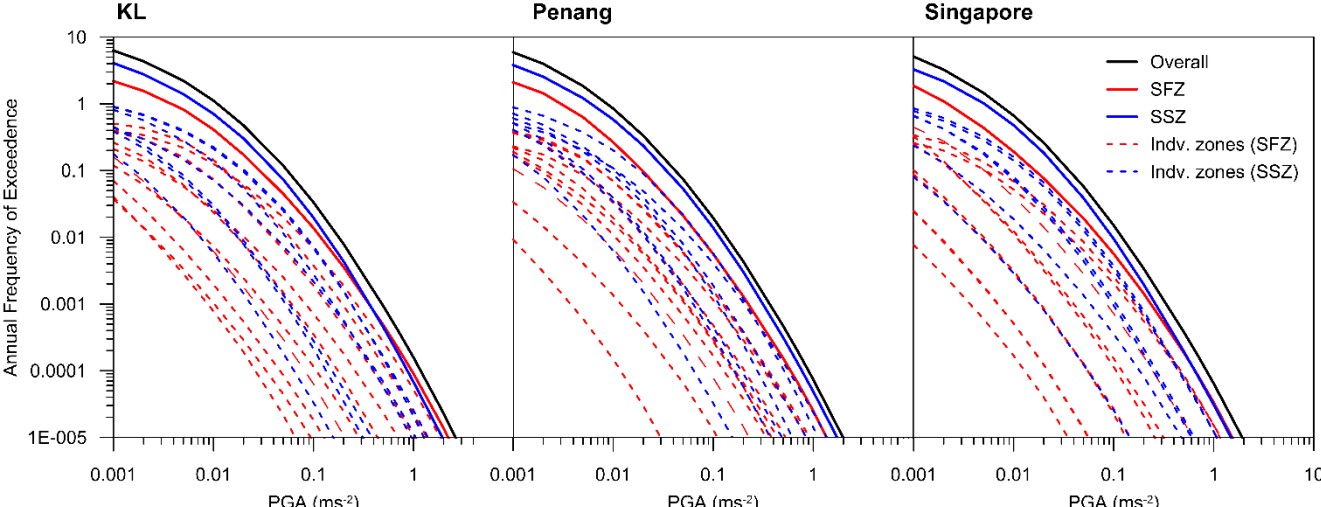

**Figure 14.** Source contribution hazard curve for KL, Penang, and Singapore.





**Table 1.** Location of MMD Seismic Stations across Peninsular Malaysia and the ground motion values recorded for the period 2004-2016 by the MMD

| Station Code | Longitude (°) | Latitude (°) | Foundation | NEHRP site classes | PGA Range (g) x $10^{-3}$ |
|---|---|---|---|---|---|
| KUM | 100.64 | 5.29 | Granite | B | 0.006 – 2.075 |
| FRM | 101.63 | 3.23 | Granite | B | 0.003 – 2.501 |
| IPM | 101.02 | 4.47 | Granite | B | 0.010 – 1.783 |
| KGM | 103.31 | 2.01 | Granite | B | 0.025 – 2.257 |
| KTM | 103.13 | 5.32 | Rock | B | 0.008 – 1.301 |
| KOM | 103.84 | 1.79 | Granite | B | 0.004 – 0.952 |
| JRM | 102.47 | 3.88 | Sandstone | B | 0.004 – 0.497 |
| PYSM_B0 | 101.68 | 2.91 | Granite | B | 0.057 – 0.939 |
| PYSM_B9 | 101.69 | 2.96 | Concrete | - | 0.173 – 2.887 |
| BKSM | 101.64 | 3.14 | Soft Soil | E | 0.099 – 2.362 |
| SASM | 101.51 | 3.09 | Soft Soil | E | 0.060 – 1.359 |
| UYSM | 101.68 | 3.27 | Soft Soil | E | 0.031 – 1.744 |
| KNSM | 101.51 | 3.27 | Soft Soil | E | 0.068 – 2.752 |
| SRSM | 101.61 | 3.36 | Rock | B | 0.072 – 6.272 |
| GTSM | 101.77 | 3.39 | Rock | B | 0.044 – 0.889 |
| JBSM | 101.86 | 3.32 | Rock | B | 0.113 – 4.362 |
| DTSM | 101.84 | 3.13 | Rock | B | 0.045 – 1.097 |
| BRSM | 101.86 | 2.90 | Rock | B | 0.046 – 1.718 |
| PJSM | 101.69 | 2.96 | Soft Soil | E | 0.066 – 1.295 |



**Table 2** List of MPEs from all three sources used in the DSHA.

| EQ no. | Date | Time (UMT) | Location | Source[a]-Country[b] | Epicentre | | Maximum magnitude[c] ($M_w$) | | | | MPE[d] ($M_w$) | Source |
|---|---|---|---|---|---|---|---|---|---|---|---|---|
| | | | | | Lat. | Long. | *1* | *2* | *3* | *4* | | |
| 1 | 14-Sep-1964 | 15:29:38 | Nicobar Island | SSZ-IND | 8.86 | 93.10 | 7.1 | 7.4 | - | - | 7.4 | USGS |
| 2 | 6-Dec-2010 | 19:26:50 | Nicobar Island | SSZ-IND | 7.88 | 91.94 | 7.5 | 7.8 | - | - | 7.8 | USGS |
| 3 | 26-Dec-2004 | 4:21:29 | Nicobar Island | SSZ-IND | 6.91 | 92.96 | 7.2 | 7.5 | - | - | 7.5 | USGS |
| 4 | 17-May-1955 | 14:49:55 | Nicobar Island | SSZ-IND | 6.82 | 93.87 | 7.0 | 7.3 | - | - | 7.3 | USGS |
| 5 | 23-Aug-1936 | 21:12:16 | Northern Sumatra | SSZ-IND | 5.32 | 94.72 | 7.0 | 7.3 | - | - | 7.3 | USGS |
| 6 | 26-Dec-2004 | 0:58:53 | Northern Sumatra | SSZ-IND | 3.29 | 95.98 | 9.1 | 9.4 | - | - | 9.4 | USGS |
| 7 | 4-Nov-2012 | 8:38:36 | Northern Sumatra | SSZ-IND | 2.33 | 93.06 | 8.6 | 8.9 | - | - | 8.9 | USGS |
| 8 | 21-Nov-1969 | 2:05:38 | Northern Sumatra | SSZ-IND | 2.00 | 94.49 | 7.6 | 7.9 | - | - | 7.9 | USGS |
| 9 | 20-Feb-1908 | 8:08:30 | Simulue | SSZ-IND | 2.77 | 95.96 | 7.4 | 7.7 | - | - | 7.7 | USGS |
| 10 | 28-Mar-1905 | 16:09:36 | Northern Sumatra | SSZ-IND | 2.09 | 97.11 | 8.6 | 8.9 | - | - | 8.9 | USGS |
| 11 | 1-Apr-1907 | 5:19:11 | Northern Sumatra | SSZ-IND | 1.87 | 94.21 | 7.8 | 8.1 | - | - | 8.1 | USGS |
| 12 | 4-Nov-1912 | 10:43:10 | Northern Sumatra | SSZ-IND | 0.80 | 92.46 | 8.2 | 8.5 | - | - | 8.5 | USGS |
| 13 | 17-Nov-1984 | 6:49:30 | Nias | SSZ-IND | 0.20 | 98.03 | 7.1 | 7.4 | - | - | 7.4 | USGS |
| 14 | 28-Dec-1935 | 2:35:31 | Kepulauan Batu | SSZ-IND | -0.29 | 98.26 | 7.6 | 8.1 | - | - | 8.1 | USGS |
| 15 | 5-Aug-1946 | 5:20:27 | Southern Sumatra | SSZ-IND | -0.75 | 99.10 | 7.3 | 7.8 | - | - | 7.8 | USGS |
| 16 | 10-Feb-1797 | - | Mentawai | SSZ-IND | -1.00 | 99.00 | 8.4 | 8.9 | - | - | 8.9 | NM87[f] |
| 17[e] | - | - | Mentawai- Siberut | SSZ-IND | -2.00 | 99.00 | 9.1 | 9.5 | - | - | 9.5 | - |
| 18 | 25-Feb-2008 | 8:36:33 | Mentawai | SSZ-IND | -2.49 | 99.97 | 7.2 | 7.7 | - | - | 7.7 | USGS |
| 19 | 25-11-1833 | - | Mentawai | SSZ-IND | -2.50 | 100.50 | 9.2 | 9.5 | - | - | 9.5 | NM87[f] |
| 20 | 25-Oct-2010 | 14:42:22 | Mentawai | SSZ-IND | -3.49 | 100.08 | 7.8 | 8.3 | - | - | 8.3 | USGS |
| 21 | 25-Jun-2014 | 19:07:25 | Southern Sumatra | SSZ-IND | -3.92 | 101.82 | 7.6 | 8.1 | - | - | 8.1 | USGS |
| 22 | 9-Dec-2007 | 11:10:26 | Southern Sumatra | SSZ-IND | -4.44 | 101.37 | 8.5 | 9.0 | - | - | 9.0 | USGS |
| 23 | 6-Apr-2000 | 16:28:26 | Southern Sumatra | SSZ-IND | -4.72 | 102.09 | 7.9 | 8.4 | - | - | 8.4 | USGS |
| 24 | 25-Sep-1931 | 5:59:52 | Southern Sumatra | SSZ-IND | -5.43 | 102.28 | 7.4 | 7.9 | - | - | 7.9 | USGS |
| 25 | 2-Mar-2016 | 12:49:48 | Southern Sumatra | SSZ-IND | -4.95 | 94.33 | 7.8 | 8.3 | - | - | 8.3 | USGS |
| 26 | 2-Apr-1964 | 1:11:50 | Seulimeum | SFZ-IND | 5.57 | 95.37 | 7.0 | 7.5 | 7.6 | 7.3 | 7.6 | USGS |
| 27 | 8-Mar-1935 | 3:58:00 | Aceh | SFZ-IND | 4.40 | 96.40 | 7.2 | 7.7 | 7.9 | 7.0 | 7.9 | USGS |
| 28 | 10-Oct-1996 | 15:21:04 | Tripa | SFZ-IND | 3.46 | 97.94 | 6.3 | 6.8 | 7.8 | 7.6 | 7.8 | USGS |
| 29 | 5-Sep-2011 | 17:55:11 | Renun | SFZ-IND | 2.97 | 97.89 | 6.7 | 7.2 | 7.9 | 7.6 | 7.9 | USGS |



| 30 | 19-May-2008 | 14:26:45 | Toru | SFZ-IND | 1.64 | 99.15 | 6.0 | 6.5 | 7.4 | 7.6 | 7.6 | USGS |
| 31 | 11-Nov-1999 | 18:05:43 | Angkola | SFZ-IND | 1.28 | 100.32 | 6.2 | 6.7 | 7.7 | 7.8 | 7.8 | USGS |
| 32 | 8-Mar-1977 | 23:17:28 | Barumun | SFZ-IND | 0.45 | 100.02 | 6.0 | 6.5 | 7.6 | 7.8 | 7.8 | USGS |
| 33 | 7-Nov-2007 | 18:37:45 | Sumpur | SFZ-IND | 0.24 | 99.96 | 6.1 | 6.6 | 6.9 | 7.8 | 7.8 | USGS |
| 34 | 6-Mar-2006 | 3:49:38 | Sianok | SFZ-IND | -0.49 | 100.50 | 6.4 | 6.9 | 7.4 | 7.8 | 7.8 | USGS |
| 35 | 9-Jun-1943 | 3:06:20 | Sumani | SFZ-IND | -0.83 | 100.74 | 7.8 | 8.0 | 7.2 | 7.8 | 8.0 | USGS |
| 36 | 19-May-1979 | 22:34:34 | Suliti | SFZ-IND | -1.08 | 100.96 | 5.4 | 5.9 | 7.4 | 7.8 | 7.8 | USGS |
| 37 | 6-Oct-1995 | 18:09:45 | Siulak | SFZ-IND | -2.05 | 101.44 | 6.8 | 7.3 | 7.3 | 7.8 | 7.8 | USGS |
| 38 | 1-Oct-2009 | 1:52:27 | Dikit | SFZ-IND | -2.48 | 101.50 | 6.6 | 7.1 | 7.2 | 7.8 | 7.8 | USGS |
| 39 | 8-Jun-1943 | 20:42:43 | Ketaun | SFZ-IND | -2.90 | 102.15 | 7.4 | 7.9 | 7.4 | 7.8 | 7.9 | USGS |
| 40 | 15-Dec-1979 | 0:02:41 | Musi | SFZ-IND | -3.30 | 102.71 | 6.6 | 7.1 | 7.3 | 7.8 | 7.8 | USGS |
| 41 | 10-Oct-1974 | 21:32:10 | Manna | SFZ-IND | -4.14 | 102.83 | 6.0 | 6.5 | 7.4 | 7.8 | 7.8 | USGS |
| 42 | 24-Jun-1933 | 21:54:49 | Kumering | SFZ-IND | -5.23 | 104.60 | 7.6 | 8.0 | 7.7 | 7.6 | 8.0 | USGS |
| 43 | 2-Apr-1919 | 0:34:59 | Semangko | SFZ-IND | -5.50 | 104.49 | 6.4 | 6.9 | 7.2 | 7.2 | 7.2 | USGS |
| 44 | 25-Oct-2000 | 9:32:23 | Sunda | SFZ-IND | -6.55 | 105.63 | 6.8 | 7.3 | 7.7 | 7.1 | 7.7 | USGS |
| 45 | 25-May-2008 | 1:36:22 | Bukit Tinggi | LI-MYS | 3.36 | 101.75 | 4.0 | 5.0 | - | - | 5.0 | MMD |
| 46 | 27-Mar-2009 | 1:46:25 | Jerantut | LI-MYS | 3.86 | 102.52 | 2.8 | - | - | - | 2.8 | MMD |
| 47 | 29-Apr-2009 | 13:53:54 | Manjung | LI-MYS | 4.15 | 100.73 | 2.4 | - | - | - | 2.4 | MMD |
| 48 | 20-Aug-2013 | 0:26:27 | Kupang (Baling) | LI-MYS | 5.59 | 100.88 | 3.8 | - | - | - | 3.8 | MMD |
| 49 | 6-Apr-1985 | 13:34:35 | Hulu Terengganu | LI-MYS | 5.10 | 102.60 | 3.8 | - | - | - | 3.8 | MMD |
| 50 | 3-Jan-2016 | 17:33:15 | Temenggor | LI-MYS | 5.55 | 101.36 | 2.8 | - | - | - | 2.8 | MMD |

[a] *Source: SSZ - Sumatran Subduction Zone; SFZ – Sumatran Fault Zone; LI – Local Intraplate*

[b] *Country: IND – Indonesia; MYS – Malaysia*

[c] *1 Maximum historical earthquake*

*2 Maximum historical earthquake + 0.3 $M_w$ for SSZ above the equator, or +0.5 $M_w$ for SSZ below the equator up to a maximum of 9.5 and SFZ until a maximum of $M_w$ 8.0, and + $M_w$ 1.0 for Bukit Tinggi*

*3 Maximum earthquake predicted from Natawidjaja & Triyoso (2007)*

*4 Maximum earthquake from Burton & Hall (2014)*

[d] *MPE : Maximum magnitude from column 1,2,3 and 4*

[e] *Event 16 is a simulated event which predicts that the Mentawai gap (0°– 2.5°S) may produce large EQ in the next few decades (Nalbant et.al 2005, Lay 2015)*

[f] *NM87: Newton and McCann (1987)*





**Table 3** Summary of locations, earthquake recurrences, and seismic activity quantification for the SSZ and SFZ

| | | Latitude (°N/°S) | Length (km) | Slip rate (mm/year) | Observation for EQ occurrence per/year past 40 years | | | | | Expected $M_wMax$ | b-value | | $R^2$ |
|---|---|---|---|---|---|---|---|---|---|---|---|---|---|
| | | | | | $4<M_w<4.9$ | $5<M_w<5.9$ | $6<M_w<6.9$ | $7<M_w<7.9$ | $8<M_w$ | | variable | fixed | |
| **Sumatran Subduction Zone** | Zone 1 | 10.00 N - 7.00 N | 342 | 44 | not utilized in this study | 3.925 | 0.725 | 0.025 | 0 | 9.0 | 0.97 | 0.83 | 0.98 |
| | Zone 2 | 7.00 N - 4.00 N | 352 | 46 | | 5.975 | 0.6 | 0.075 | 0 | 9.1 | 1.06 | 0.83 | 0.99 |
| | Zone 3 | 4.00 N - 2.00 N | 311 | 50 | | 4.85 | 0.875 | 0 | 0.075 | 9.2 | 0.56 | 0.83 | 0.97 |
| | Zone 4 | 2.00 N - 0.00 | 278 | 56 | | 4.375 | 0.375 | 0.225 | 0.025 | 9.3 | 0.76 | 0.83 | 0.98 |
| | Zone 5 | 0.00 - 2.00 S | 265 | 56 | | 2.875 | 0.775 | 0.1 | 0 | 9.4 | 0.72 | 0.83 | 0.99 |
| | Zone 6 | 2.00 S - 4.00 S | 278 | 59 | | 3.15 | 0.725 | 0.15 | 0 | 9.5 | 0.68 | 0.83 | 0.98 |
| | Zone 7 | 4.00 S - 7.00 S | 448 | 62 | | 13.825 | 1.775 | 0 | 0.025 | 9.5 | 0.85 | 0.83 | 0.99 |
| | Overall | 13.00 N - 7.00 S | 2274 | 44-62 | | 38.975 | 5.850 | 0.575 | 0.125 | - | 0.83 | - | 0.99 |
| **Sumatran Fault Zone** | Zone 1 | 5.57 N - 5.01 N | 82 | 13 | 0.75 | 0.125 | 0.05 | 0.025 | no available records | 7.6 | 0.55 | 0.84 | 0.97 |
| | Zone 2 | 5.01 N - 4.71 N | 50 | 27 | 1.725 | 0.225 | 0.075 | 0 | | 7.6 | 0.84 | 0.84 | 0.97 |
| | Zone 3 | 4.71 N - 4.45 N | 45 | 27 | 1.925 | 0.3 | 0.025 | 0 | | 7.9 | 1.13 | 0.84 | 0.96 |
| | Zone 4 | 4.45 N - 3.99 N | 83 | 27 | 0.975 | 0.225 | 0.025 | 0.025 | | 7.9 | 0.65 | 0.84 | 0.93 |
| | Zone 5 | 3.99 N - 3.16 N | 142 | 27 | 1.6 | 0.325 | 0.075 | 0.025 | | 7.9 | 0.67 | 0.84 | 0.99 |
| | Zone 6 | 3.16 N - 2.23 N | 136 | 27 | 2.35 | 0.375 | 0.1 | 0 | | 7.9 | 0.8 | 0.84 | 0.99 |
| | Zone 7 | 2.23 N - 1.18 N | 138 | 27 | 3.5 | 0.525 | 0.05 | 0 | | 7.6 | 1.06 | 0.84 | 0.98 |
| | Zone 8 | 1.18 N - 0.27 S | 182 | 26 | 2.4 | 0.375 | 0.075 | 0 | | 7.8 | 0.89 | 0.84 | 0.97 |
| | Zone 9 | 0.27 S - 1.71 S | 194 | 28 | 1.2 | 0.175 | 0.1 | 0.025 | | 8.0 | 0.53 | 0.84 | 0.96 |
| | Zone 10 | 1.71 S - 3.09 S | 191 | 23 | 2.55 | 0.575 | 0.1 | 0.025 | | 7.8 | 0.7 | 0.84 | 0.98 |
| | Zone 11 | 3.09 S - 4.34 S | 196 | 13 | 2.15 | 1.05 | 0.1 | 0 | | 7.9 | 0.91 | 0.84 | 0.97 |
| | Zone 12 | 4.34 S - 5.29 S | 141 | 11 | 0.925 | 0.725 | 0.075 | 0.025 | | 8.0 | 0.64 | 0.84 | 0.97 |
| | Zone 13 | 5.29 S - 6.00 S | 90 | 11 | 1.45 | 0.725 | 0.075 | 0 | | 7.7 | 0.84 | 0.84 | 0.94 |
| | Overall | 5.57 N - 6.00 S | 1670 | 11-28 | 23.5 | 5.725 | 0.925 | 0.15 | | | 0.84 | - | 0.99 |

