# Peer review of "Revisiting Seismic Hazard Assessment for Peninsular Malaysia Using Deterministic and Probabilistic Approaches"

_Natural Hazards and Earth System Sciences, 2018_

## Referee Comment (RC1) · Anonymous Referee #1 · 20 Mar 2018

**General Comments**

I found this paper to be a thorough, thoughtful, comprehensive study of seismic hazard assessment in Peninsular Malaysia. The scientific merit of this paper is excellent, with many angles considered. The paper itself is very well-written and the work clearly and concisely explained, which is much appreciated from the perspective of a reviewer/reader. The paper is a synthesis of an impressive amount of work, one which seems it could even be presented in 2 – 3 papers instead of 1.

This manuscript will, I think, be an important contribution for both scientific literature, and hazard assessment in the region. I recommend that it be published with minor

revisions. I do not have any major criticisms for the scientific contents of the paper. My main comments pertain mostly to some of the figures, and a few comments regarding the GMPE section of the paper, and this is the only reason I select minor revisions instead of technical corrections. Please find below my specific comments, as well as technical comments. Line reference as follows: P.#, L.#.

**Specific Comments**

1) P.6, L.1: You mention the local intraplate earthquakes and faults, and plot them on a couple figures. If possible, describe the type of faults these are (strike-slip, normal, thrust, etc.), as this is important for future studies to consider (with regards to directivity, hanging-wall effects, etc.), and I think also important for readers to understand if and how any of these effects have been considered in the GMPEs, later on in the manuscript.

2) P.12: I think it is important for readers to know a little more about the GMPEs, such as what are the basic components of the functional forms (i.e., are there hanging wall effects and other more detailed effects, or just magnitude/magnitude squared/geometric spreading/intrinsic attenuation terms?) What do the attenuation parameters look like, and how does that compare to attenuation in the region (if there are studies of Q here)? How is the site represented – is it basic NEHRP classes in all of these GMPEs? How do the models compare to each other? I don't think this has to be a long discussion, but as the rest of the paper is so comprehensive I don't think an extra paragraph or two here describing the ground-motion models could hurt, as they are a significant component of seismic hazard assessment.

3) P.13, L.19: I noticed in several places in the paper (including this line), the authors mention that they use "mean" values from the ground-motion models. Generally, ground-motion models predict median ground-motion – are the models you are using instead predicting the mean? If so, I encourage you to perhaps add some text in the discussion discussing the implications of this (i.e., it can sometimes inflate the hazard

as opposed to using the median value).

4) Figure 1: I found this figure a little difficult to interpret. I appreciate that it is required to pack a lot of information into it, so I have a few suggestions that could include all this material, and make it a little easier to interpret: a. Make the coastlines thicker, and/or color the land/water separately b. Add some longitude/latitude tick marks and/or grid lines to the figure, to orient the reader and help them understand what the map projection is c. Perhaps add topography or bathymetry? (Though this could make it busier, and harder to read) d. Place a box around the approximate area/location of Figures 2,3,4 and 8,9,10, and 12 since I think they are slightly different from what I can tell e. Perhaps code the intraplate faults based on the type, and/or add direction of motion f. Place direction of motion on the SFZ g. Caption: Add a citation for the intraplate fault database.

5) Figure 2: A few comments - a. The label "Mantel" should be "Mantle" b. Add the direction of motion of the SFZ c. It is a little hard to see the Local Intraplate label on the top right – I would suggest adding this to the caption that is already on the bottom left, with "major seismic activities" and "subducting direction" d. Caption: Describe the diverging white arrows; I am assuming the numbers ("approx.. 10km", "> 2000km") are thicknesses, but I would suggest explicitly writing this in the caption; Add a citation for the intraplate faults, like Figure 1.

6) Figure 4: There are a few things that made this figure a little difficult for me to interpret, I have a few suggestions: a. Add some arrows indicating which boxes are Zone 1,2,3,4 b. Perhaps reduce the opacity on the SSZ and SFZ zoned areas, as it is hard to see the background seismicity through them c. It is a little hard to see the text for Zones 5 – 7 in the SSZ, maybe make this text white, or put an opaque gray box behind all these zone texts? d. Not a major comment, small, but the last portion of the M in KM is cut off on the scale, bottom right

7) Figures 8 and 9: I have a few suggestions – most of them are in the interest of

making the figures more similar to Figure 12, in the interest of being able to directly compare the results of the DHA vs. PSHA. a. Making the city labels a little larger, it is hard to see them b. Make the coastlines, geographic regions lines a little thicker, hard to see c. Make the fault labels a little larger d. Perhaps color the ocean like in Figure 12, for consistency? e. Add gridlines on the plot, like in Figure 12

**Technical Corrections**

1) In abstract, P.1, L.14-15: Perhaps also give these PGA values in percent g? For example, "PGAs of 0.07 – 0.80 m/s2 (0.7 – 8.1 percent g)..."

2) P.2, L.28 – 29: "This method, nonetheless, is not free of criticism as studies have observed that PSHA is merely a numerical creation with a hazy mathematical concept and the use of it may lead to risky or overly conservative engineering design". Perhaps a bit nit-picky, but I feel this is a bit harsh on PSHA, and a subjective statement. The main criticism of PSHA is that it cannot be validated, and therefore I do not think its criticism can "observe" that it is numerical creation, or has a hazy mathematical concept... but perhaps these studies can "suggest" that is mathematical, and has challenges in validation due to lack of data. I still contend, however, that its mathematical concept is not hazy...probabilities are not hazy, they are used in major financial decisions every day and are at the root of most capitalistic endeavors, and those who apply these "hazy" mathematical concepts seem to profit from them...just as an example.

3) A purely stylistic suggestion, of course authors' choice: The introduction is very well laid out, and has a decent amount of background. My only suggestion would be to place the main study focus before the description of PSHA, i.e., put the material from P.2, L.31 through P.3, L.9 before the discussion of DHA vs. PSHA, which could then motivate this discussion.

4) P.5, L.17 – 18: "Lying dextral and parallel about 200km away from the trench to accommodate the oblique convergence along the plate margin is the Sumatran Fault Zone. This 1900 km long strike-slip fault..." I found myself a little confused about

whether the fault was dextral in its motion, or if dextral referred to its position; perhaps change to: "lying east and parallel... This 1900km long dextral strike-slip..." ?

5) P.6, L.19: In the paragraph preceding this line, perhaps reference Figure 1 or 2, to indicate where the reader can find the local intraplate faults on a map.

6) P. 8, L.9: "within the same grid in the past.". I am assuming "in the past" refers to since 1797, as described on L.4 – if I am correct, perhaps add that in? "within the same grid since 1797"

7) P.11,L.8-9: I am assuming the b-value was computed on events with M> 4.0 (SFZ) and 5.0 (SSZ) because of the network's magnitude of completeness? It looks like on Figure 5, the event start to fall off here. If I am correct, perhaps state that here to clarify.

8) P.14, L.8: I suggest changing "local intraplate earthquakes" here to LI earthquakes, since you have an abbreviation for it.

9) Figure 3: I do not see any of the LI greater than M 3.0 events (pink dots) – should there be any?

10) Figure 6: In the caption, perhaps describe what the recorded data shown is from (dates, etc.); Add a goodness of fit of the GMPEs to the data, if you have them?

11) Figure 7: Is Beta-value (column heading) supposed to be b-value?

12) Figure 8: Caption – is "mean" GMPE supposed to be median here?

13) Figure 12: Add the study abbreviations (A06, A05, etc.) into the caption.

14) Table 1: Is PYSM_B9 the site located on a building, which you said was not included in the study? If so, perhaps add an asterisk in the table and caption to specify.

Please also note the supplement to this comment:
https://www.nat-hazards-earth-syst-sci-discuss.net/nhess-2018-51/nhess-2018-51-RC1-supplement.pdf

---

## Referee Comment (RC2) · Anonymous Referee #2 · 29 Mar 2018

This is a revision for the paper entitled "Revisiting Seismic Hazard Assessment For Peninsular Malaysia Using Deterministic And Probabilistic Approaches" By Daniel Weijie Loi et al. The paper is well organized but a major revision is required before publication. Following are my main points to be considered: 1- The used catalog is not subjected to completeness analysis, therefore, the authors consider the earthquakes are complete for the entire magnitude range (4-9.1) along the catalog period (1907-2016). I am highly skeptic about this. Please provide, at least, a completeness analysis showing that earthquakes of magnitude 4.0 are complete along the entire period. 2- The catalog shows no earthquakes generated by the local intraplate faults. Really I do not know if these faults are active or inactive one to be included or excluded from

the calculations. Please follow the following points: a- Provide evidences for the activity of all mapped major intraplate faults. b- Define their dimension and rate of slip along each of them. c- Define the associated maximum magnitude and recurrence interval based upon the above data. If these faults are active, then the seismic hazard will change dramatically. Using the maximum recorded PGA values is not the proper way for seismic hazard assessment. 3- Bases for subdividing SSZ into 7 areas and line seismic source zones are unclear and very confusing. Sumatra earthquake 2004 initiated at latitude near 3.2 degree N and extended for about 1200 km northward till about 14 degree N, rupturing at least zones 3, 2 and 1. These zone were ruptured in one earthquake, therefore, I found it strange to subdivide it into three different seismic zones. Segments 4, 5, and 6 have almost the same slip rate, thus their segmentation is questionable for me. Generally segmentation along SSZ is unclear, therefore, geological, tectonic, seismological evidences should be provide to support the current segmentation. 4- According to Wells and Coppersmith, 1994, Strasser et al., 2010, and Blaser et al., 2010, all the provided fault lengths can not produce the expected magnitudes in Table 3. 5- Gutenberg-Richter (1944) approach to define b-values imposes the unrealistic assumption that the maximum potential earthquake is unbounded and unrelated to the seismotectonic setting. Therefore, I prefer to use the truncated exponential model instead of G-R (1944) model, which contradicts the idea of maximum magnitude as it is open from its both ends. 6- Figure 5 shows a very strange piece of data, where the logarithm of the cumulative annual frequency for earthquakes with magnitude 9.1 is Zero, meaning that the annual frequency of this range of magnitude is 1.0. Actually we do not have an earthquake with magnitude 9.1 or larger every years in this area. A great mistake is committed and should be reconsidered. Authors seem to use the same recurrence parameters for both area and line sources. Please use rate of slip to define the recurrence parameters for the fault sources. But firat authors should show how did they calculate the slip rate and show whether their calculations contain creep components or not and show whether the time span for calculation the slip rate is representative or not. Comparison of the results using the area and line sources

should be provided. 7- According to Figure 5, the maximum observed magnitude at zones 1, 2 is less than 7.6 (1.5 magnitude unit less than the maximum magnitude assigned for these seismic zones). Please comment. Such inconsistency is observed at many other regions. The solution is to combined the provided segmented seismic sources into proper larger ones. 8- Local intraplate faults and the seismic activity at Sabah are not included in the PSHA. 9- The distances employed in the Ground Motion Prediction Equations (GMPE) is the hypocentral distance as indicated in figure 6. This kind of distances considers the earthquake as a point and can not be used for earthquakes that cause ruptures up to 1200 km. Even it can not be used for local source that can produce earthquakes of magnitude 5.0. Recent GMPE avoid using the hypocentral distance as it overestimates the distance. Although the authors used local GMPE, but It is not appropriate for the current use. I suggest to use Rrup or Rjb within appropriate GMPE for the studied area. Please always provide more details about the used GMPE (e.g. minimum amd maximum distance for applicability, type of horizontal ground motion used, tectonic environment, magnitude used, shear-wave velocity, etc.). Of most important is to define the standard deviation for the used GMPE 10- GMPE used seems not to calculate the ground motion in terms of response spectra, which are the most important input parameters for engineers, especially if they are asked to use the IBC codes. PGA is OK if the Euro code is to be applied, but it is just an isolated value on the time history and neither represents the ground motion nor correlates well with the damage potential of shaking. I highly recommend to provide hazard maps in terms of short period and 1.0 sec spectral period for the two return periods (475 and 2475 years) in addition to the PGA maps. 11- The main advantage of the PSHA is the combination of all magnitudes, distances, and effects. Thus all seismic sources that might affect the area of interest should be included in each single run. Separation of SSZ and SFZ in the logic tree is an mistake as it underestimate the seismic hazard. of course, different seismic source models can be used, but in each model all the seismic sources should be used in each single run. For example authors may consider each of SSz and STZ as single or more in one branch of the logic tree while the their preferable

source model is on the other branch. Segmentation of the seismic zone into area and lines zones is acceptable.

MINOR REVISIONS a- Page 2 line 13: unclear b- Page 2 line 20: use scenario instead of value c- Page 3 line 2: add (.) after 2016) d- Page 3 line 20: Provide the magnitude of Sabah earthquakes e- Page 6 line 32: the velocity range should be changed f- Page 10 line 17: It means that data are incomplete for some magnitude range for period 1907-1977. As mentioned above, completeness analysis is a must. g- Page 10 line 23: only if the slip does not has a creep component. Please comment on the creep component in the total slip if any. Figure 1: 1- Some green colors are shown in western Malysia, please correct 2- Symbole of SSZ is inconsistent with the figure 3- The figure should show coordinates, scale and North direction Table 3: 1- Boundary conditions should be modified as the following example (7<Mw<7.9, 7.9$\geq$Mw and so on) Results are not discussed as substantial modification is required before getting the right results

---

## Referee Comment (RC3) · C.-H. Chan (Referee) · 3 Apr 2018

This paper assesses seismic hazard for Peninsular Malaysia by considering both deterministic and probabilistic approaches. I found the importance of this work since few studies focusing on the seismic hazard of this region and this study could provide a better assessment. However, I have a number of questions about the paper. Below I detail my comments and questions.

Major questions:

DSHA and PSHA: Usually the hazard level determined by DSHA should be higher

than or equal to that by PSHA since DSHA considers characteristic events regardless it occurrence probability. Thus, I am surprised that the DSHA results (Figures 8 and 9) has significant lower hazard than the PSHA ones (Figure 12 b). I am confused how it could happen. I wish authors could have a good explanation for it.

Catalogue completeness: Implementing an incomplete catalogue could result in over-estimation of earthquake recurrence for large magnitude. In this study, earthquakes with M≥4.0 since 1907 (or 1976, stated in Line 15 of Page 10) are implemented. However, the catalogue incompleteness is shown in Figure 5b that seismicity with M≤4.2 does not follow the G-R law, resulting in a lower-b-value (shown in Table 3, since it is uncommon having b-value smaller that 0.8, especially in active tectonic environments). A G-R model with a low b-value expect higher occurrence rate for large magnitude and higher hazard.

Fault parameters: The fault parameters (e.g., segmentation, maximum magnitude, slip rate) implemented in this study are obtained from previous researches. These parameters, however, sometimes are different from the Indonesian Hazard Map (the 2010 version can be download through: https://www.google.com.sg/url?sa=t&rct=j&q=&esrc=s&source=web&cd=1&cad=rja&uact=8&ved=0ahUKEwiTnLyC6qjVAl As5RkrlohnA, updated version has been proposed in 2017). For example, the slip rate of the Sumatran Fault implemented in this study (Lines 19-23 of Page 5) is significant higher than those proposed by the Indonesian Hazard Map; segmentation of the Sumatran fault is different. If authors prefer the current setting, some description on the discrepancy between each other is required.

Logic tree branch: Since occurrences of earthquakes with different magnitudes are independent to each other, it is not necessary to be implemented into logic tree (as described in Line 32 of Page 12 and Line 1 of Page 13).

Point source for DSHA: An earthquake could be regarded as a point source when its magnitude is related small, whereas a line or plan source should be implemented for

a large event. Experience (in the form of scaling law) suggests fault length could be longer than 10 km for an M≥6.0 event. Besides, for DSHA of the Bukit Tinggi Fault, the epicenter of a coming event is controversial. Thus, I would suggest conducting a series of scenario considering different rupture lines along the fault and report the highest shaking level for each calculation node (, suggesting the worst case).

Miscellaneous questions and comments:

Some of the references in the references list cannot be found through the internet (e.g., Loi et al., 2016; Loi et al., submitted). It makes audience difficult to evaluate the credibility of this study. Thus, I would suggest detailed description of the referred studies in the text (e.g., credibility of implemented GMPEs).

I feel this study tries to link with design code, thus I would suggest to assess seismic hazard not only in peak ground acceleration, but also spectral acceleration.

Line 4 of Page 4: 'activity' instead of 'recurrence'?

Line 8 of Page 4 and Figure 1: Coordinates are expected in Figure 1 so audience can understand the region described in the text.

Lines 29-30 of Page 5: A locking depth of 15 km is implemented, while the Indonesian Hazard Map utilized 20 km. Although I do not expect significant difference in the results, I am looking forward to an explanation or a reference for this parameter.

Line 31 of Page 5: An unnecessary comma should be removed.

Line 32 of Page 6: Site class E is soft soil, whereas Vs30 ranging from 760 to 1500 ms-1 is defined as site A.

Line 25 of Page 13: 'times' instead of 'fold'?

Lines 12 and 18 of Page 14 and Figure 8: Location of KL should be denoted in Figure 8.

Figure 1: Do orange lines denote active faults? If so, please specify their reference(s). Besides, I am confused on the alignments of 'Tectonic plates boundary'. For the West of Sumatra as example, I expect the boundary should be further to the west (fit the alignment of the Sunda Trench).

Figure 2: What is the meaning of '>2000 km' in the figure? Thickness of Mantle, or the depth of the boundary between crust and mantle? Besides, there is a typo for 'Mantle'.

Figure 3: Some events took place at the West of the Sunda Trench should not belong to the Sumatran subduction zone.

Table 3: Although the epicenter of the 2004 M9.1 event is in Zone2, part of its rupture zone locates on Zone 1. Thus I suggest MwMax of 9.1 (or even 9.2) for Zone 1.

Thus, I suggest this manuscript can be published after a major revision.

Chung-Han Chan, Nanyang Technological University, Singapore, April, 2018.

---

## Author Comment (AC1) · 9 Jun 2018

**RC 1**

| Reviewer comments | Author response |
|---|---|
| *1) P.6, L.1: You mention the local intraplate earthquakes and faults, and plot them on a couple figures. If possible, describe the type of faults these are (strike-slip, normal, thrust, etc.), as this is important for future studies to consider (with regards to directivity, hanging-wall effects, etc.), and I think also important for readers to understand if and how any of these effects have been considered in the GMPEs, later on in the manuscript.* | Based on the information available from the literature and the geological map of peninsular Malaysia, the intraplate faults are normal and strike-slip faults. This information will be added in the paper. Due to our limited access to the detailed information, the effects mentioned by the Reviewer were not considered in the GMPEs. |
| *2) P.12: I think it is important for readers to know a little more about the GMPEs, such as what are the basic components of the functional forms (i.e., are there hanging wall effects and other more detailed effects, or just magnitude/magnitude squared/geometric spreading/intrinsic attenuation terms?) What do the attenuation parameters look like, and how does that compare to attenuation in the region (if there are studies of Q here)? How is the site represented – is it basic NEHRP classes in all of these GMPEs? How do the models compare to each other? I don't think this has to be a long discussion, but as the rest of the paper is so comprehensive I don't think an extra paragraph or two here describing the ground-motion models could hurt, as they are a significant component of seismic hazard assessment.* | Considering that each GMPE was developed independently by different researchers, providing more details for every GMPE utilized in the current study will inflate the size of the present paper. However, we acknowledge that it is important for readers to at least have a quick understanding of the GMPEs. We will, therefore, include a table with information about the GMPEs used for SHA. The information provided will include the regions, tectonic settings, magnitude ranges, distances, functional forms, and standard deviations of all the GMPEs. |
| *3) P.13, L.19: I noticed in several places in the paper (including this line), the authors mention that they use "mean" values from the ground-motion models. Generally, ground-motion models predict median ground-motion – are the models you are using instead predicting the mean? If so, I encourage you to perhaps add some text in the discussion discussing the implications of this (i.e., it can sometimes inflate the hazard as opposed to using the median value).* | Yes we are using mean values (not the median values). Majority of GMPEs listed in John Douglas' GMPE compendium (see http://www.gmpe.org.uk/) deals with the mean values. Strasser et al. (2008) also discuss why ground-motion residual distribution being generally assumed to be normal with a mean of zero and a standard deviation σ. |
| *4) Figure 1: I found this figure a little difficult to interpret. I appreciate that it is required to pack a lot of information into it, so I have a few suggestions that could include all this material, and make it a little easier to interpret:*
*a. Make the coastlines thicker, and/or color the land/water separately* | We thank the Reviewer for the suggestions to improve the clarity of the Figure. We accept suggestions a, b, and d. Adding topography/bathymetry (suggestion c.) make the map even busier. We prefer to stick with the current map (from ArcGIS Esri. 2015) and make slight modifications to it. |

| | |
|---|---|
| *b. Add some longitude/latitude tick marks and/or grid lines to the figure, to orient the reader and help them understand what the map projection is*
*c. Perhaps add topography or bathymetry? (Though this could make it busier, and harder to read)*
*d. Place a box around the approximate area/location of Figures 2,3,4 and 8,9,10, and 12 since I think they are slightly different from what I can tell*
*e. Perhaps code the intraplate faults based on the type, and/or add direction of motion*
*f. Place direction of motion on the SFZ g. Caption: Add a citation for the intraplate fault database* | The intraplate fault lines (suggestion 4e.) are digitized lines obtained from the Geological Map of Peninsular Malaysia (2014). It would be extremely small if there were to be added in an already crowded map. These lines are clearly shown in the blown up figures of the peninsula (Figures 8 and 9). |
| *5) Figure 2: A few comments –*
*a. The label "Mantel" should be "Mantle"*
*b. Add the direction of motion of the SFZ*
*c. It is a little hard to see the Local Intraplate label on the top right – I would suggest adding this to the caption that is already on the bottom left, with "major seismic activities" and "subducting direction"*
*d. Caption: Describe the diverging white arrows; I am assuming the numbers ("approx.. 10km", "> 2000km") are thicknesses, but I would suggest explicitly writing this in the caption; Add a citation for the intraplate faults, like Figure 1.* | We thank the Reviewer for the suggestions. All points will be taken onboard in the revised version. |
| *6) Figure 4: There are a few things that made this figure a little difficult for me to interpret, I have a few suggestions:*
*a. Add some arrows indicating which boxes are Zone 1,2,3,4*
*b. Perhaps reduce the opacity on the SSZ and SFZ zoned areas, as it is hard to see the background seismicity through them*
*c. It is a little hard to see the text for Zones 5 – 7 in the SSZ, maybe make this text white, or put an opaque gray box behind all these zone texts?*
*d. Not a major comment, small, but the last portion of the M in KM is cut off on the scale, bottom right.* | We thank the Reviewer again. All suggestion will be considered in the revision. |
| *7) Figures 8 and 9: I have a few suggestions – most of them are in the interest of making the figures more similar to Figure 12, in the interest of being able to directly compare the results of the DHA vs. PSHA.*
*a. Making the city labels a little larger, it is hard to see them* | All the suggestions, barring d, will help us improve the figures and will be considered in revising the figures. It is quite difficult to color the ocean as the base map was obtained from ArcMap 10.4. Hence, we will have to find another ocean map and overlay the other information which may lead to some discrepancies especially |

| | |
|---|---|
| *b. Make the coastlines, geographic regions lines a little thicker, hard to see*
*c. Make the fault labels a little larger 3*
*d. Perhaps color the ocean like in Figure 12, for consistency?*
*e. Add gridlines on the plot, like in Figure 12.* | in terms of boundaries. Nevertheless, we will attempt to make the coastlines and geographic region lines thicker to address this problem. |
| ***Specific comments*** | |
| *1) In abstract, P.1, L.14-15: Perhaps also give these PGA values in percent g? For example, "PGAs of 0.07 – 0.80 m/s2 (0.7 – 8.1 percent g)…"* | We prefer to stick with ms$^{-2}$ for consistency throughout the text. However, we are happy to add values in percent g if required by the journal. |
| *2) P.2, L.28 – 29: "This method, nonetheless, is not free of criticism as studies have observed that PSHA is merely a numerical creation with a hazy mathematical concept and the use of it may lead to risky or overly conservative engineering design". Perhaps a bit nit-picky, but I feel this is a bit harsh on PSHA, and a subjective statement. The main criticism of PSHA is that it cannot be validated, and therefore I do not think its criticism can "observe" that it is numerical creation, or has a hazy mathematical concept… but perhaps these studies can "suggest" that is mathematical, and has challenges in validation due to lack of data. I still contend, however, that its mathematical concept is not hazy…probabilities are not hazy, they are used in major financial decisions every day and are at the root of most capitalistic endeavors, and those who apply these "hazy" mathematical concepts seem to profit from them…just as an example.* | Although we have paraphrased the above statement from literature, we appreciate the Reviewer's view and we will reword the statements as to be non-controversial. |
| *3) A purely stylistic suggestion, of course authors' choice: The introduction is very well laid out, and has a decent amount of background. My only suggestion would be to place the main study focus before the description of PSHA, i.e., put the material from P.2, L.31 through P.3, L.9 before the discussion of DHA vs. PSHA, which could then motivate this discussion.* | We thank the Reviewer for the suggestion. We have added a paragraph in the revision and it reads much better now. |
| *4) P.5, L.17 – 18: "Lying dextral and parallel about 200km away from the trench to accommodate the oblique convergence along the plate margin is the Sumatran Fault Zone. This 1900 km long strike-slip fault…" I found myself a little confused about whether the fault was dextral in its motion, or if dextral referred to its position; perhaps change to: "lying east and parallel… This 1900km long dextral strike-slip…" ?* | We will reword the sentences to avoid confusion as suggested by the Reviewer. |

| | |
|---|---|
| *5) P.6, L.19: In the paragraph preceding this line, perhaps reference Figure 1 or 2, to indicate where the reader can find the local intraplate faults on a map.* | This is done in the revision. |
| *6) P. 8, L.9: "within the same grid in the past.". I am assuming "in the past" refers to since 1797, as described on L.4 – if I am correct, perhaps add that in? "within the same grid since 1797"* | We will reword accordingly in the revised manuscript. |
| *7) P.11,L.8-9: I am assuming the b-value was computed on events with M> 4.0 (SFZ) and 5.0 (SSZ) because of the network's magnitude of completeness? It looks like on Figure 5, the event start to fall off here. If I am correct, perhaps state that here to clarify.* | The use of the entire magnitude range (4.0 – 9.1) was initially considered based on the observation that earthquakes causing felt ground motion in the peninsula start at $M_w$ 4.0. We therefore assumed that the catalog is complete. However, taking into account that both Reviewer #2 and Reviewer #3 have noted that the completeness analysis is essential for PSHA, we have re-performed a completeness analysis using the Stepp (1972) method and include it in the revised manuscript. |
| *8) P.14, L.8: I suggest changing "local intraplate earthquakes" here to LI earthquakes, since you have an abbreviation for it.* | Thank you. Changed in the revised manuscript. |
| *9) Figure 3: I do not see any of the LI greater than M 3.0 events (pink dots) – should there be any?* | Perhaps it is confusing. We will revise Figure 3 to make these (LI events of Mw >3.0) more distinct. |
| *10) Figure 6: In the caption, perhaps describe what the recorded data shown is from (dates, etc.); Add a goodness of fit of the GMPEs to the data, if you have them?* | We are not showing the recorded data, etc. in the current paper as these data have been used in a separate manuscript (submitted). Having the exact same data/details will increase the size of this paper and possibly lead to plagiarism accusation. |
| *11) Figure 7: Is Beta-value (column heading) supposed to be b-value?* | Thank you for alerting us to this typographic error. This is supposed to be b-value and will be revised accordingly. |
| *12) Figure 8: Caption – is "mean" GMPE supposed to be median here?* | It is meant to be "mean." |
| *13) Figure 12: Add the study abbreviations (A06, A05, etc.) into the caption.* | These will be added in the revised manuscript. Thank you. |
| *14) Table 1: Is PYSM_B9 the site located on a building, which you said was not included in the study? If so, perhaps add an asterisk in the table and caption to specify.* | This will be added in the revised manuscript. |

---

## Author Comment (AC2) · 9 Jun 2018

**RC 2**

P: Page
L: Line

| Reviewer comments | Author response |
|---|---|
| *1- The used catalog is not subjected to completeness analysis, therefore, the authors consider the earthquakes are complete for the entire magnitude range (4-9.1) along the catalog period (1907- 2016). I am highly skeptic about this. Please provide, at least, a completeness analysis showing that earthquakes of magnitude 4.0 are complete along the entire period.* | The use of the entire magnitude range (4.0 – 9.1) was initially considered based on the observation that earthquakes causing felt ground motion in the peninsula starts at $M_w$ 4.0. We, therefore, assumed that the catalog is complete. However, taking into account that both Reviewer #2 and Reviewer #3 have noted that the completeness analysis is essential for PSHA, we have re-performed a completeness analyses using the Stepp (1972) method and will be included in the revised manuscript. |
| *The catalog shows no earthquakes generated by the local intraplate faults. Really I do not know if these faults are active or inactive one to be included or excluded from the calculations. Please follow the following points:* | The local intraplate earthquakes have been inactive in the past, but Shuib (2009) noted that due to the massive 2004 Aceh earthquake, some of the local intraplate faults may have been reactivated. This was evident in a series of mini earthquakes felt between 2008 and 2009 at Bukit Tinggi and several other areas within the peninsula. These events were recorded by the MMD and are tabulated in Table 2 (nos. 45-50). Therefore, while these faults are not completely active, they are also not absolutely inactive. |
| *a- Provide evidences for the activity of all mapped major intraplate faults.* | Due to their relative inactiveness, limited information (slip rate etc.) is available to date from past literature, the Malaysian Meteorological Department (MMD) and Department of Mineral of Geosciences on the definition of the activity rates within the local intraplate. Amongst the local intraplate faults, only the Bukit Tinggi fault has been studied more closely by local researchers (Shuib, 2009; Shuib et al., 2017) revealing that there are several likely active faults in Peninsular Malaysia based on earthquake epicenter distribution. These geomorphologic studies, however, did not indicate how "active" these intraplate faults are.
The mapping of major and minor faults lines were digitized from the Geological Map by MMD (2014). |
| *b- Define their dimension and rate of slip along each of them.* | As mentioned above in 2a, this is not possible with limited information. |
| *c- Define the associated maximum magnitude and recurrence interval based upon the above data. If these faults are active, then the seismic hazard will change dramatically. Using the maximum recorded PGA values is not the proper way for seismic hazard assessment.* | Apart from the Bukit Tinggi event that we have modelled at a high magnitude of 6.0 (Mw) due to concerns raised by Looi et al. (2003) – P14, L13, the maximum magnitude for all the remaining 5 local intraplate earthquakes were based on the records provided by MMD.  Only DSHA was performed for these events, PSHA was NOT |

| | conducted as we were not able to calculate the recurrence interval based on the limited available information.

The maximum recorded PGA values were not utilized in the SHA but shown for comparison of suitability of the GMPEs to be used in the SHA.

*Note: It should be noted that due to the limited geological information available and the relative inactiveness of the faults, PSHA for the local intraplate faults was NOT conducted. Only DSHA was conducted using point sources. This has been highlighted in P12, L13. Nevertheless, we will make it clearer in the revision.* |
|---|---|
| *3- Bases for subdividing SSZ into 7 areas and line seismic source zones are unclear and very confusing. Sumatra earthquake 2004 initiated at latitude near 3.2 degree N and extended for about 1200 km northward till about 14 degree N, rupturing at least zones 3, 2 and 1. These zone were ruptured in one earthquake, therefore, I found it strange to subdivide it into three different seismic zones. Segments 4, 5, and 6 have almost the same slip rate, thus their segmentation is questionable for me. Generally segmentation along SSZ is unclear, therefore, geological, tectonic, seismological evidences should be provide to support the current segmentation.* | We are aware that various researchers have segmented the SSZ differently using different geological and tectonic methods of segmentation in the past. For example, Hanus (1996) demarcated 30 zones across the Sumatran subduction and fault zones based on earthquake foci, Franke et al. (2008) performed digital imaging based on the 2004 to 2005 massive Sumatra earthquake, and Petersen et al. (2007) conducted SHA based on deep and shallow events. However, we are not aware that there is a clear segmentation that defines the whole of 4000+ km long SSZ that can be precisely defined/segmented/modelled for PSHA. Therefore, we have modelled the SSZ based on the seismological evidence in the subduction zone by dividing them into various subdivisions at 2-3° latitudinal intervals to avoid overlap of zones when PSHA analyses is conducted. With earthquake rupture dimension being different for each independent event, there is no exact methodology to segment the length of each sub-division.

We have, therefore, modelled the individual zones of SSZ at 2-3° intervals as rupture length of large earthquakes from past events approximately within this range, as shown in Subarya et al. (2006)
- 2005 earthquake (Mw 8.6) – appox. 2.5°N to 0°
- 2007 earthquake (Mw 8.4) – appox. 3.0°S to 5.0°S
- 1833 earthquake (Mw 9.0)- approx. 2.5°S to 5.6°S

As for the question raised by Reviewer #2 on what happens during an exceptional event such as the 2004 earthquake which ruptured for an extensive length, we have considered another model that takes into account the entire subduction length as |

| | part of the logic tree, and this will be added in the revised manuscript. |
|---|---|
| *4- According to Wells and Coppersmith, 1994, Strasser et al., 2010, and Blaser et al., 2010, all the provided fault lengths cannot produce the expected magnitudes in Table 3.* | Expected $M_w$MAx was not based on calculation of the fault length and depth for SSZ, rather consideration that an earthquake of such a high magnitude is possible (P10, L9-12).

As for SFZ, these values were extracted from literature in Table 2, with an upper boundary assigned. – (P8, L23-30).

As the word "expected" may be confusing, we will revise it to "modelled." |
| *5- Gutenberg-Richter (1944) approach to define b-values imposes the unrealistic assumption that the maximum potential earthquake is unbounded and unrelated to the seismotectonic setting. Therefore, I prefer to use the truncated exponential model instead of G-R (1944) model, which contradicts the idea of maximum magnitude as it is open from its both ends.* | We appreciate the Reviewer's comments on the model choice. We may point out that the G-R method has been used in other recent published work also (e.g., Ullah et al. 2015, Wang et al. 2016). We have conducted our analysis based on what we understand best and believe that we have obtained sensible results using the G-R method. |
| *6- Figure 5 shows a very strange piece of data, where the logarithm of the cumulative annual frequency for earthquakes with magnitude 9.1 is Zero, meaning that the annual frequency of this range of magnitude is 1.0. Actually we do not have an earthquake with magnitude 9.1 or larger every years in this area. A great mistake is committed and should be reconsidered. Authors seem to use the same recurrence parameters for both area and line sources.* | We thank the Reviewer for pointing out this mistake. The label was supposed to be "cumulative frequency" instead of "cumulative annual frequency". The label in Figure 5 will be corrected accordingly. |
| *Please use rate of slip to define the recurrence parameters for the fault sources. But first authors should show how did they calculate the slip rate and show whether their calculations contain creep components or not and show whether the time span for calculation the slip rate is representative or not.*
*Comparison of the results using the area and line sources should be provided.* | The slip rates for both the SSZ and SFZ were not calculated. They were obtained from literature as mentioned in Section 3 (see P5, L4-7). We are hence unable to comment on the creep component calculation. It should be noted that different slip rates for the SSZ have been reported in the literature. However, majority of the literature agrees that the slip rate increases from north to south along the subduction line. |
| *7- According to Figure 5, the maximum observed magnitude at zones 1, 2 is less than 7.6 (1.5 magnitude unit less than the maximum magnitude assigned for these seismic zones). Please comment. Such inconsistency is observed at many other regions. The solution is to combined the provided segmented seismic sources into proper larger ones.* | While Figure 5 does show the observed magnitude at various zones to be lower than the maximum magnitude assigned, the expected $M_w$Max utilized for the PSHA was once again not based solely on the historical values.
The expected MwMax values for zones in both the SSZ and SFZ were modelled to be as high as 9.5 and 8.0, respectively, because we intended to model them as the worst-possible case scenario. Considering that the 2004 earthquake was able rupture >1200 km and produce an earthquake of 9.1$M_w$, we wonder why is it not possible for it produce a similar magnitude rupture again in the future? |

| | The upper boundary of the expected $M_w$Max though differs from zone to zone. The modelled values are explained on P10, L8-14. |
|---|---|
| *8- Local intraplate faults and the seismic activity at Sabah are not included in the PSHA.* | Sabah does not fall within the scope of the current study; only peninsular Malaysia is only considered. We will revise Fig. 1 to give a clearer representation of our study area. |
| *9- The distances employed in the Ground Motion Prediction Equations (GMPE) is the hypocentral distance as indicated in figure 6. This kind of distances considers the earthquake as a point and cannot be used for earthquakes that cause ruptures up to 1200 km. Even it cannot be used for local source that can produce earthquakes of magnitude 5.0. Recent GMPE avoid using the hypocentral distance as it overestimates the distance. Although the authors used local GMPE, but it is not appropriate for the current use. I suggest to use Rrup or Rjb within appropriate GMPE for the studied area.* | We appreciate the Reviewer's suggestion for use of alternative parameters. However, the available information on the rupture plane is limited. Therefore, we would prefer to stick with hypocentral distances. Moreover, with distances as long as 1200 km, the effects of using various distance parameters (Repi, Rhyp, Rrup, or Rjb) for this region are not huge, as also noted by Van et al. (2016). The GMPEs utilized for DSHA and PSHA (SSZL18, SFZL18, S16 and SM00) were mainly derived based on the hypocentral distances, and therefore, we have conducted the analyses based on Rhyp. |
| *Please always provide more details about the used GMPE (e.g. minimum amd maximum distance for applicability, type of horizontal ground motion used, tectonic environment, magnitude used, shear-wave velocity, etc.). Of most important is to define the standard deviation for the used GMPE.* | More details regarding the GMPEs including the standard deviations of the parameters that have were used for the DSHA and PSHA in this work will be provided in the form of a Table in the revised manuscript. |
| *10- GMPE used seems not to calculate the ground motion in terms of response spectra, which are the most important input parameters for engineers, especially if they are asked to use the IBC codes. PGA is OK if the Euro code is to be applied, but it is just an isolated value on the time history and neither represents the ground motion nor correlates well with the damage potential of shaking. I highly recommend to provide hazard maps in terms of short period and 1.0 sec spectral period for the two return periods (475 and 2475 years) in addition to the PGA maps.* | We appreciate the Reviewer's recommendation and acknowledge that the response spectra are an important input parameter for engineers. We have already acknowledged the limitation of the present work. Some of the GMPEs (LSSZ18, LSFZ18 and SM00) used in this work do not include the coefficients required to calculate the response spectra. Hence, we have omitted them from the current work. The reason why we have focused our work on the PGA at bedrock is because as recently as 2016, the Department of Standards Malaysia have drafted a seismic resistance design code based on the Eurocode 8 which specify the notional design of PGA at bedrock. |
| *11- The main advantage of the PSHA is the combination of all magnitudes, distances, and effects. Thus all seismic sources that might affect the area of interest should be included in each single run. Separation of SSZ and SFZ in the logic tree is an mistake as it underestimate the seismic hazard. of course, different seismic source models can be used, but in each model all the seismic sources should be used in each single run. For example authors may consider each of SSz and STZ as single or more in one branch of the logic tree while the their preferable source model is on the other branch. Segmentation of the seismic zone into area and lines zones is acceptable.* | We thank the Reviewer for picking up this mistake. It was an oversight from us in separating the two different source models in the logic tree. We have already repeated the analyses by combining all the related seismic sources in a single run. The logic tree branch in Figure 7 will also be redrawn with the new results. |

**References**

Franke, Dieter, et al. (2008) "The great Sumatra–Andaman earthquakes—Imaging the boundary between the ruptures of the great 2004 and 2005 earthquakes." *Earth and Planetary Science Letters* 269.1-2: 118-130.

Hanuš, V., A. Špičák, and J. Vaněk. (1996) "Sumatran segment of the Indonesian subduction zone: morphology of the Wadati-Benioff zone and seismotectonic pattern of the continental wedge." *Journal of Southeast Asian Earth Sciences* 13.1: 39-60

Mustaffa Kamal Shuib, Mohammad Abdul Manap, Felix Tongkul, Ismail Bin Abd Rahim, Tajul Anuar Jamaludin, Noraini Surip5, Rabieahtul Abu Bakar, Mohd Rozaidi Che Abas, Roziah Che Musa, Zahid Ahmad. (2017) "Active Faults In Peninsular Malaysia With Emphasis On Active Geomorphic Features Of Bukit Tinggi Region." Malaysian Journal Geosciences (MJG) 1(1) (2017) 13-26

Subarya, Cecep, et al. (2006) "Plate-boundary deformation associated with the great Sumatra–Andaman earthquake." *Nature* 440.7080: 46.

Ullah, S., Bindi, D., Pilz, M., Danciu, L., Weatherill, G., Zuccolo, E., Issuk, A., Mikhailova, N., Abdrakhmatov, K. and Parolai, S. (2015) "Probabilistic seismic hazard assessment for Central Asia." *Ann. Geophys.*, *58*(1), p.0103S.

Wang, Y.J., Chan, C.H., Lee, Y.T., Ma, K.F., Shyu, J.H., Rau, R.J. and Cheng, C.T. (2016) "Probabilistic seismic hazard assessments for Taiwan." *Terr. Atmos. Ocean. Sci.*, *27*(3), pp.325-340.

---

## Author Comment (AC3) · 9 Jun 2018

**RC 3**

| Reviewer comments | Author response |
|---|---|
| *DSHA and PSHA: Usually the hazard level determined by DSHA should be higher than or equal to that by PSHA since DSHA considers characteristic events regardless it occurrence probability. Thus, I am surprised that the DSHA results (Figures 8 and 9) has significant lower hazard than the PSHA ones (Figure 12 b). I am confused how it could happen. I wish authors could have a good explanation for it.* | The reason for the hazard map based on DSHA has lower values compared to that from PSHA is that DSHA was modelled based on point sources from historical events while PSHA was modelled using line and areal sources. Hence, while some points in DSHA as tabulated in Table 3 and Figure 8 may occur at a large magnitude within similar zones to those in PSHA, these events are located further from the site when compared to the areal and line models in PSHA in Figure 4. |
| *Catalogue completeness: Implementing an incomplete catalogue could result in overestimation of earthquake recurrence for large magnitude. In this study, earthquakes with M≥4.0 since 1907 (or 1976, stated in Line 15 of Page 10) are implemented. However, the catalogue incompleteness is shown in Figure 5b that seismicity with M≤4.2 does not follow the G-R law, resulting in a lower-b-value (shown in Table 3, since it is uncommon having b-value smaller that 0.8, especially in active tectonic environments). A G-R model with a low b-value expect higher occurrence rate for large magnitude and higher hazard.* | The use of the entire magnitude range (4.0 – 9.1) was initially considered based on the observation that earthquakes causing felt ground motion in the peninsula start at $M_w$ 4.0. We, therefore, assumed that the catalog is complete. However, taking into account that both Reviewer #2 and Reviewer #3 have noted that the completeness analysis is essential for the PSHA, we have already performed a completeness analyses using the Stepp (1972) method and the results will be included in the revised manuscript.

Although it is quite uncommon for b-value to be smaller than 0.8, previous literature (Petersen et al. 2007, Pailoplee and Choonwong 2014, and Pailoplee 2017) showed that the b-value in this region can be relatively low in some cases. With our new completeness analysis results we will report revised b-values (together with their standard deviation) in Table 3 in the revised manuscript. |
| *Fault parameters: The fault parameters (e.g., segmentation, maximum magnitude, slip rate) implemented in this study are obtained from previous researches. These parameters, however, sometimes are different from the Indonesian Hazard Map (the 2010 version can be download through:, updated version has been proposed in 2017). For example, the slip rate of the Sumatran Fault implemented in this study (Lines 19-23 of Page 5) is significant higher than those proposed by the Indonesian Hazard Map; segmentation of the Sumatran fault is different. If authors prefer the current setting, some description on the discrepancy between each other is required.* | We have explained the reason for why we prefer the segmentation suggested by Burton & Hall (2014) compared to Natawidjaja & Triyoso (2007) in page 10, lines 8 – 15 of the original manuscript. As for the slip rates, these values were not provided by Burton & Hall (2014). We have, therefore, extracted the slip rate values from Natawidjaja & Triyoso (2007). For example, Zone 1 in Burton & Hall (2014) is approximately the same as Seulimium fault in Natawidjaja & Triyoso (2007). Hence, the slip rate of 13mm/year reported in Natawidjaja & Triyoso (2007) was adopted and input into the zonation suggested by Burton and Hall (2014). |

| | We will give a brief explanation of the values in the revised manuscript. |
|---|---|
| *Logic tree branch: Since occurrences of earthquakes with different magnitudes are independent to each other, it is not necessary to be implemented into logic tree (as described in Line 32 of Page 12 and Line 1 of Page 13).* | Taking into consideration the mistake made in conducting the PSHA as pointed out by Reviewer #2, we have amended our logic tree. We have redone the PSHA using line and areal sources for both the Sumatran subduction and Sumatran fault. The revised logic tree structure will be included in the revised manuscript. |
| *Point source for DSHA: An earthquake could be regarded as a point source when its magnitude is related small, whereas a line or plan source should be implemented for a large event. Experience (in the form of scaling law) suggests fault length could be longer than 10 km for an M≥6.0 event. Besides, for DSHA of the Bukit Tinggi Fault, the epicenter of a coming event is controversial. Thus, I would suggest conducting a series of scenario considering different rupture lines along the fault and report the highest shaking level for each calculation node (, suggesting the worst case).* | We thank the Reviewer for the suggestion and appreciate his/her expertise on this. However, literature has shown that point sources have been conducted at relatively high magnitudes. For example, Kolathayar et al. (2012), and Orozova and Suhadolc (1999) have performed their DSHA using point sources at higher magnitude. Although the Reviewer could be right in terms of better representation using line or areal sources, our intention was to conduct the DSHA based on the location of past historical events scaled to an upper boundary magnitude limitation.

For the point source at the Bukit Tinggi event, the epicenter was modelled at the current point based on the data provided by the Malaysian Meteorological Department (MMD). Although a series of mini-earthquakes did occur close to the point of reference (3.36°N, 101.75°E), the event at (3.36°N, 101.75°E) was the largest. That is the reason why we have chosen this particular point as our point source. As we cannot pinpoint the exact location of the next earthquake along this source, our intention was to perform a critical scenario with a reasonably high magnitude that has been scaled up based on past events. |
| *Some of the references in the references list cannot be found through the internet (e.g., Loi et al., 2016; Loi et al., submitted). It makes audience difficult to evaluate the credibility of this study. Thus, I would suggest detailed description of the referred studies in the text (e.g., credibility of implemented GMPEs).* | Condensed information regarding the GMPEs together with their respective standard deviations (the subject matter of a manuscript currently under consideration by another journal) will be provided in the form of a table in the revised manuscript. We are also happy to provide the unpublished manuscript for a perusal by the Reviewers of this journal. |
| *I feel this study tries to link with design code, thus I would suggest to assess seismic hazard not only in peak ground acceleration, but also spectral acceleration.* | Some of the GMPEs (SSZL18, SFZL18 and SM00) utilized in this work do not include the coefficients required to calculate the response spectra. Hence, we have omitted them from the |

| | current work. Clearly, this is a limitation of the present work as also noted in the manuscript. Our work focused on the PGA at bedrock because as recently as 2016, the Department of Standards Malaysia have drafted a seismic resistance design code based on the Eurocode 8 which specifies the notional design of PGA at bedrock. |
|---|---|
| *Line 4 of Page 4: 'activity' instead of 'recurrence'?* | Thank you. We will rectify this in revised manuscript. |
| *Line 8 of Page 4 and Figure 1: Coordinates are expected in Figure 1 so audience can understand the region described in the text.* | We will include coordinates in the revised manuscript. |
| *Lines 29-30 of Page 5: A locking depth of 15 km is implemented, while the Indonesian Hazard Map utilized 20 km. Although I do not expect significant difference in the results, I am looking forward to an explanation or a reference for this parameter.* | We have not calculated this value, but extracted from Natawidjaja & Triyoso (2007) as mentioned in page 5 line 30-31. |
| *Line 31 of Page 5: An unnecessary comma should be removed.* | We will correct this in the revised manuscript. |
| *Line 32 of Page 6: Site class E is soft soil, whereas Vs30 ranging from 760 to 1500ms-1 is defined as site A.* | It was a typographical error, and we thank the Reviewer for pointing it out. Site E should be Vs30 of less than 180ms-1. We will correct this in the revised manuscript. |
| *Line 25 of Page 13: 'times' instead of 'fold'?* | We will correct this in the revised manuscript. |
| *Lines 12 and 18 of Page 14 and Figure 8: Location of KL should be denoted in Figure 8.* | We will denote location of KL in the revised manuscript. |
| *Figure 1: Do orange lines denote active faults? If so, please specify their reference(s). Besides, I am confused on the alignments of 'Tectonic plate boundary'. For the West of Sumatra as example, I expect the boundary should be further to the west (fit the alignment of the Sunda Trench).* | Figure 1 will be modified accordingly. However, for the alignments and fault lines, the base source was obtained from ArcGIS Desktop Esri (2015), and has been referenced in Figure 1. |
| *Figure 2: What is the meaning of '>2000 km' in the figure? Thickness of Mantle, or the depth of the boundary between crust and mantle? Besides, there is a typo for 'Mantle'.* | Thickness of mantle. Will correct this in the revised manuscript. |
| *Figure 3: Some events took place at the West of the Sunda Trench should not belong to the Sumatran subduction zone.* | Although tectonically they may not belong to the SSZ, we have considered them as part of SSZ because these events were large enough to cause ground motion felt in Peninsular Malaysia. Thus, instead of modelling them altogether as a different model/region, we have considered and modelled them under SSZ. |
| *Table 3: Although the epicenter of the 2004 M9.1 event is in Zone 2, part of its rupture zone locates* | We appreciate the Reviewer's suggestion. The revised PSHA has already considered this and the results will included in the revised paper. |

| | |
|---|---|
| *on Zone 1. Thus I suggest MwMax of 9.1 (or even 9.2) for Zone 1.* | |
| *Thus, I suggest this manuscript can be published after a major revision.* | We thank the Reviewer for the valuable comments that has improved our paper. We appreciate the Reviewer's recommendation for publication after we have satisfactorily answered the queries and concerns. |

**References**

Kolathayar, S., Sitharam, T.G. and Vipin, K.S., 2012. Deterministic seismic hazard macrozonation of India. *Journal of earth system science*, *121*(5), pp.1351-1364.

Pailoplee, S., 2017. Probabilities of Earthquake Occurrences along the Sumatra-Andaman Subduction Zone. *Open Geosciences*, *9*(1), pp.53-60.

Pailoplee, S. and Choowong, M., 2014. Earthquake frequency-magnitude distribution and fractal dimension in mainland Southeast Asia. *Earth, Planets and Space*, *66*(1), p.8.

Orozova, I.M. and Suhadolc, P., 1999. A deterministic–probabilistic approach for seismic hazard assessment. *Tectonophysics*, *312*(2-4), pp.191-202.